# Study of Fault Identification of Clearance in Cam Mechanism

**Xuefang Chang \*** , **Hongxia Pan, Jian Xu and Tong Wang**

College of Mechatronic Engineering, North University of China, Taiyuan 030051, China;
19810192@nuc.edu.cn (H.P.); 20050106@nuc.edu.cn (J.X.); w1984202910@163.com (T.W.)
\* Correspondence: 20080005@nuc.edu.cn

**Abstract:** The clearance between the main roller and the cylinder cam has an important impact on the operational status of the bolt. In order to achieve the clearance and recoil of the cylinder cam mechanism, a bolt model was built and studied. The leading dynamic roller group and the dynamic bolt were analyzed. Positive stress and friction force change quickly on the high-speed leading roller group, influencing multiple aspects. When the main roller suddenly rotates or reverses in several short intervals, the assembling clearance and the massive friction force between the main roller and the cam curve slot are the main wear factors in theoretical analysis and test experiments. The leading roller group, the pure rolling criterion of the cylindrical cam mechanism of the automaton, is derived in this paper. Moreover, a new fault diagnosis method is developed, based on the Variational Box Dimension Kernel Fuzzy Mean Clustering Algorithm (VK) algorithm, which combines the variational mode decomposition and fractal box-counting dimension (VMD-FBCD) and the fuzzy clustering algorithms with a kernelized Mahalanobis distance (KMD-FC). With the simulation validation from 2810 samples, diagnosis using the VK algorithm is found to have a higher accuracy rate and slightly lower fallacy rate and to be faster than other diagnostic methods of similar studies. The results in terms of the recoil cylinder cam mechanism's multi-body dynamic analysis and high-speed fault identification also have significance as a reference for solving similar problems.

**Keywords:** cam mechanism; assembling clearance; pure rolling; fault identification

## 1. Introduction

Much research on the profile of the cam mechanism has been conducted, changing the base circle radius, the eccentricity of the cam, and other parameters to obtain the required motion conversion and energy output. Moreover, optimizing the parameter settings further optimizes the cam mechanism's achievable functions. At present, this design concept has been widely used in the fields of transportation and aerospace. Through the study of the parabolic-cam-roller–quasi-zero-stiffness (PCR-QZS) isolator and the derivation of the theoretical formula, the conditions under which the parabolic cam roller (PCR) is superior to the circular cam roller (CCR) are obtained. Finally, a vibration isolator with a lower stiffness and transfer coefficient on a broader area near the equilibrium position is achieved [1]. The idea has gradually begun to prevail, of which followers are applying a multi-degree-of-freedom system to suppress the nonlinear dynamic motion. This system includes adding a spring-damping system at the end of the mechanism to turn the system into a multi-degree-of-freedom system, and using a fast Fourier transform through an infrared camera to reduce the corresponding frequency and peaks of the roller follower nonlinearity [2]. Based on the three-involute profile of the cam, the gap between the roller and the guide rail, the contact position of the roller and the cam curve groove, the dynamic change between the cam and the square groove key, the change in the Lyapunov exponent, and, from the rigid body, the change in contact stress are analyzed from a dynamic point of view. The Newton–Euler variance is used for analytical calculation. Finally, it is concluded that, the greater the contact stress, the greater the Lyapunov exponent and the greater the bending deflection value [3]. By studying the influence of guideway size and cam,

angular velocity changes in the nonlinear dynamic response of the system, power spectrum analysis, phase plane mapping, and maximum Lyapunov exponent parameters are used to study the periodic motion of non-followers. Finally, the nonlinear response peaks of the roller follower at the four positions of 0°, 90°, 180°, and 270° [4] are investigated. Multiple degrees of freedom (spring-damper system) are added at the end of the follower stem, and the peak of nonlinear response is reduced by 20.89%, 47.76%, 58.2%, and 67.16%, respectively [5]. In addition, the optimized design structure of the intermediate connection of three rollers is added between the cam and the typical follower, which can generate multiple contact points between the cam and the follower at any time. Experiments show that the contact stress is 30–47% lower than that of the traditional system [6]. In order to investigate the nonlinear dynamic effect of assembly clearance on the system when recoiled, Xu et al. from NUC of China conducted a dynamic analysis of the main roller from the perspective of system dynamics and deduced the necessary conditions for pure rolling when the main roller suddenly reversed [7].

In order to study the influence of the contact load on the bending deformation of the cam profile, the bending deflection of the disc cam with the roller follower was analyzed under a constant angular velocity of the cam. Solidworks and Ansys software were used to measure the follower's deflection. The cam profile's dynamic response and bending deformation were studied and analyzed. Finally, the bending deflections of path 1, path 2, and path 3 were reduced by 73.425%, 85.925%, and 61.467%, respectively [8]. Another hot spot in this field in recent years has been the analysis and life prediction of the lubricating environment of the cam-roller follower system. By establishing the electro-hydro-dynamic (EHL) model, and providing the model with three basic parameters, the minimum oil film thickness, the maximum oil film pressure, and the minimum oil film thickness distribution, it is possible to analyze the kinematic parameters of the entire system, such as the velocity, acceleration, and displacement of the cam, under actual working conditions through numerical equations. The lubrication conditions of the mechanism are estimated to predict the actual life [9]. Considering non-Newtonian effects, thermal effects, and variable angular velocity angles of the rollers from the perspective of mixed lubrication, using load distribution and surface roughness analysis, a quasi-static analysis of the heavy-duty cam roller follower was carried out [9,10]. The results show that, when the non-Newtonian effect and thermal effect are ignored when the contact surface is in a low-friction state, the main roller is in a pure rolling state, and the lubrication performance is also better. Furthermore, as the friction in the needle roller contact is upgraded, the contact force, non-Newtonian properties, and viscosity-pressure properties are also upgraded, and the SRR increases sharply. After that, the system changes from the full-film lubrication state to the mixed lubrication state [10].

The cam mechanism constitutes the heavy-duty part of the automaton of the Gatling gun, and it plays a significant role with respect to the shot speed, such as shot frequency, shot accuracy, and shot intensity. The traditional design of the main roller and cam focuses on reasonably optimizing the bolt's motion law and rationally arranging a cyclic diagram of the device's law [11]. A new optimization design method for the end face of the cam curve groove was recommended to reduce the lateral impact of the main roller [12]. The pure rolling criterion of the main roller for the fixed cam followed with rollers was deduced theoretically [13]. The power spectrum analysis, the phase-plane mapping, and the Lyapunov exponent parameters were involved in investigating the non-periodic motion of the roller follower [5]. The profile errors were revealed by changing the center distance between the cam and the pivots with the clearance [14]. A numerical analysis method for identifying the assembly defects of the cam curved groove mechanism was proposed following the Hertz contact theory [15]. The multi-body dynamic analysis and the optimization of the end groove curve among the rollers were performed [16,17]. However, the face-to-face interaction between the fixed cam curve groove and the roller and bolt group with a high speed has been less studied, especially failure diagnosis and prediction on the roller-cam

device. The positive pressure and friction force change quickly and influence multiple aspects of the high-speed revolving main roller. Intensive further study is needed.

This study introduces a fixed cam device with several rollers. Additionally, the requirements for the pure roller of a cylindrical cam in recoil with clearance are investigated. Figure 1 illustrates the automaton mechanism [7]. The structure of the fixed-cam curve slot and the bolt is shown in Figure 2. The specific structure of the bolt body is shown in Figure 3.

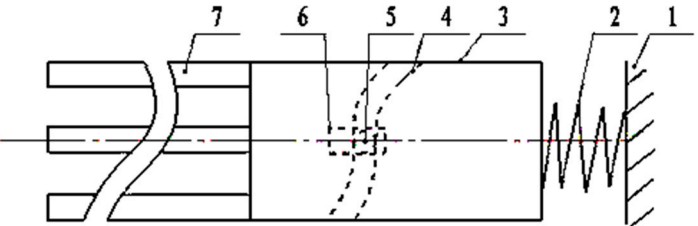

**Figure 1.** Cam curve slot mechanism and structure of the working parts. Note: 1—mechanism carriage, 2—buffer, 3—box mechanism, 4—fixed cam curve slot, 5—main roller, 6—working parts, and 7—barrels.

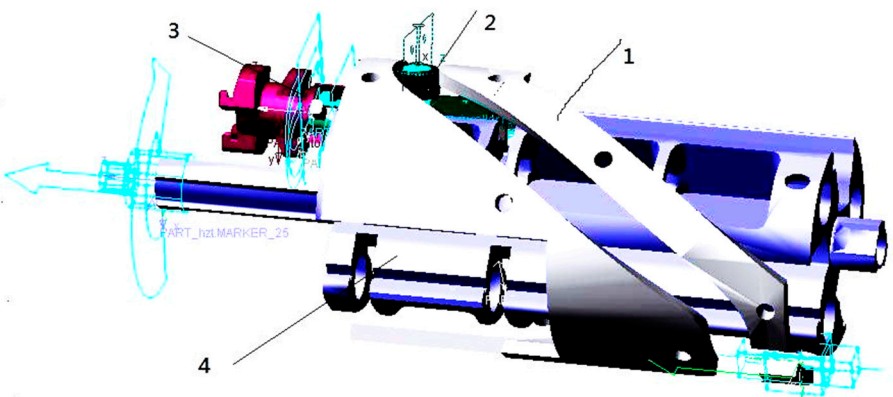

**Figure 2.** Cam transmission mechanism. Note: 1—fixed cam curve slot, 2—main roller, 3—bolt group, and 4—revolving body.

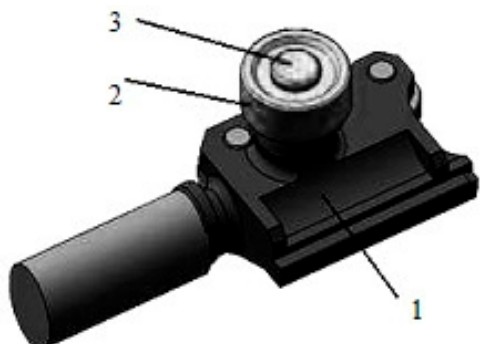

**Figure 3.** Bolt. Note: 1—bolt body, 2—main roller, and 3—the rolling axis.

## 2. Primary Roller Modeling and Dynamics Analysis

In this mechanism, the cam is co-axially assembled with a revolving body. In the duty status, the revolving body rotates along its axis. The revolving body, which is the fourth part as shown in Figure 2, has longitudinal grooves, which is the first part as shown in Figure 2, on its cross-section. They are co-axially assembled. The bolt group also has guide rails matching the guide groove on the revolving body. The bolt group, the third part as shown in Figure 2 and the first part as shown in Figure 3, revolves around the revolving



body in synchronization, and it transfers along the guide groove on the revolving body. The bolt group is co-axially connected to every main roller. The rollers, the second part as shown in Figures 2 and 3, run along with the curve slot on the cam, which is the first part as shown in Figure 2 and the fourth part as shown in Figure 1. The distinguished merit of the device is that the designers can place several sets of guide grooves and bolt groups along the circumferential direction as needed to achieve high efficiency.

The bolt group works under a high-speed and heavy-load transfer motion mechanism. It is a critical part of the cam design, and the cam curve slot determines its rule of motion. The diagram of the transfer motion mechanism is shown in Figure 1. A rigorous analysis of the force status of the main roller needs to consider its motion status. It denotes the rotary inertia of the roller relative to its own rotary axis as $J$ and the mass of the whole bolt group as $m$. Generally, in the cam mechanism design and the pure rolling state of the main roller, the radius of the cam groove curvature, the size of the motion distortion, and the rate of change in the contact strength are critical factors. Under high speed and heavy load, it has been pointed out that the revolving body's acceleration of the Gatling weapon, driven by the external energy, should decrease [13]. The driven method is helpful for pure rolling and reduces the normal pressure. As for the energy-driven method, the automaton driven by external energy is better than that driven by internal and external energy.

When the cam roller mechanism has a gap and is operated at a high speed and with a heavy load, the rotation rate can reach 200–1200 rpm. Its acceleration can reach 2000–6000 m/s$^2$, and the translational velocity can reach 10–30 m/s. The angular speed can reach 100–150 rad/s. Under the rotary longitudinal guide rail, the reciprocating motion of the bolt is given as unity. The greater the acceleration of the movement group, the greater the force of the main roller. Pure rolling and accidental side impact occur under the guidance of the cam groove, and its angular speed can reach nearly 3000 rad/s. The cam groove can support 27,500 N, and the bolt displacement can reach 10–15 m/s. The model is shown in Figures 4 and 5 below.

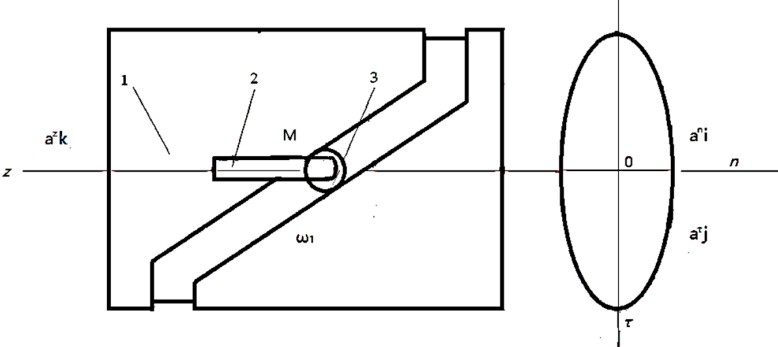

**Figure 4.** Model of the cam-roller follower mechanism. 1—fixed cylindrical cam, 2—bolt body, and 3—main roller.

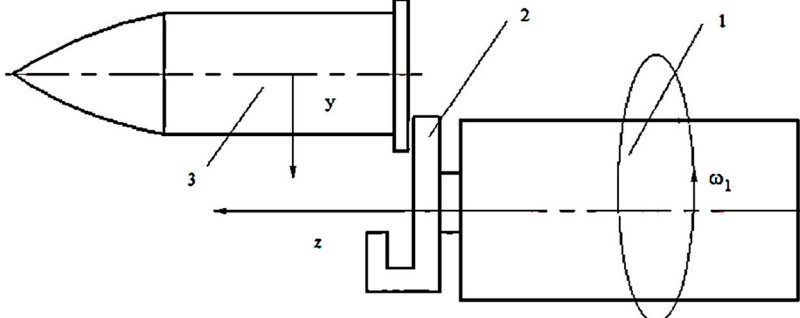

**Figure 5.** Bolt body model. Note: 1—bolt body, 2—bolt nose, and 3—a missile.

As the cycles of the main roller slide along the cam curve slot surface, the friction condition deteriorates gradually, and the driving power consumption increases. Meanwhile, friction sintering is caused on the contact surface, leading to mechanical malfunction or motion distortion. The dynamic characteristic of the roller group in the pure rolling status is the key to reducing the mechanism wear and driving power consumption.

*2.1. Main Roller Kinematics Analysis*

The velocity decomposition of the main roller in a cam curve slot is illustrated in Figure 6.

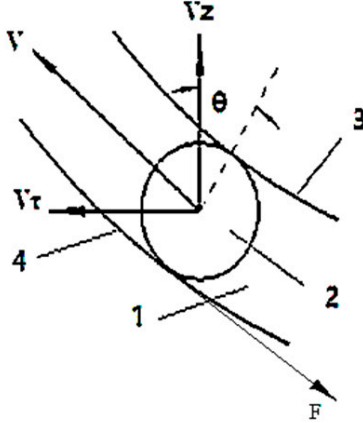

**Figure 6.** Velocity decomposition of the main roller with no clearance. Note: 1—cam curve slot, 2—main roller, F—the friction force, 3—the rear-end surface, and 4—the front-end surface.

For the K roller group, the pure rolling of the mass transitional velocity matrix V consists of K rows and K columns:

$$V = \frac{V_z}{\sin \theta} \qquad (1)$$

where, $\theta_i$ is the motion pressure angle of the roller ($\mathrm{diag}(\theta) = [\theta_1, \theta_2, \theta_3, \cdots, \theta_n]$) and $V_z$ is the longitudinal velocity vector of the bolt. If every main roller is pure rolling, the roller group angular velocity matrix $\omega$ can be represented as:

$$\omega = \frac{V}{R} \qquad (2)$$

where R is the radius of the main roller, $\varepsilon$ is the angular acceleration of the main roller, and it can be represented as:

$$\varepsilon = \frac{\dot{V}_z}{R}\psi(\theta) + \frac{V_z}{R}\dot{\psi}(\theta) \qquad (3)$$

where

$$\psi(\theta) = \csc \theta, \psi'(\theta) = -\psi(\theta)\varphi(\theta)\dot{\theta} \qquad (4)$$

The variation $V_z$ with the angular velocity of the revolving body $\omega_1$ is expressed as follows:

$$V_z = i\omega_1 \qquad (5)$$

where the letter i represents the transmission ratio of the revolving body to the bolt body and can be determined by the cam curve design. The acceleration of the bolt body is expressed as follows:

$$\frac{dV_z}{dt} = \omega_1{}^2\frac{di}{d\phi} + i\alpha \qquad (6)$$

where $\phi$ is the angle of revolution of the revolving body, and $\alpha$ is the angular acceleration of the revolving body. The rate of change in the pressure angle matrix $\theta$ is expressed as follows:

$$\frac{d\theta}{dt} = \omega\dot{\theta} \tag{7}$$

By substituting Equations (5) and (6) into Equation (3), the roller angular acceleration can be represented as

$$\varepsilon = \frac{i\omega_1{}^2\psi(\theta)}{R}\left[\frac{i}{i} + \frac{\alpha}{\omega_1{}^2} - \varphi(\theta)\dot{\theta}\right] \tag{8}$$

where

$$\varphi(\theta) = \cot(\theta) \tag{9}$$

Based on Equation (7), with the given cam groove curve and the angular velocity $\omega_1$ of the revolving body, the angular acceleration matrix $\varepsilon$ of the roller group can be obtained under the assumption of pure rolling.

The inertia force of the main roller is expressed as follows:

$$F_g = m\cdot(\omega_1{}^2\frac{di}{d\phi} + i\alpha). \tag{10}$$

The inertia moment $M_g$ is expressed as follows:

$$M_g = \frac{iJ\omega_1{}^2\psi(\theta)}{R}\left[i./i + \alpha/\omega_1{}^2 - \varphi(\theta)\dot{\theta}\right] \tag{11}$$

### 2.2. Main Roller Pure Rolling Analysis

The main roller connected with one bolt body is taken as the study subject. In the pure rolling status, the forces imposed on the main roller consist of a static friction force *Ff*, a vertical contact force *N*, and two constraint forces, *Tr* and *Tz*, on the roller axis. Whether the roller slides or rolls relative to the pin axis or not, the friction circle radius $\rho_0$ of the roller axis is approximately 0.1 to 0.001 times the radius of the pin roll Rn. It is a small quantity relative to the arm of force equal to the main roller radius R, and it is often approximated to zero in the calculation process. Supposing a static friction coefficient *fs*, the simplified acting force analysis is illustrated in Figure 7.

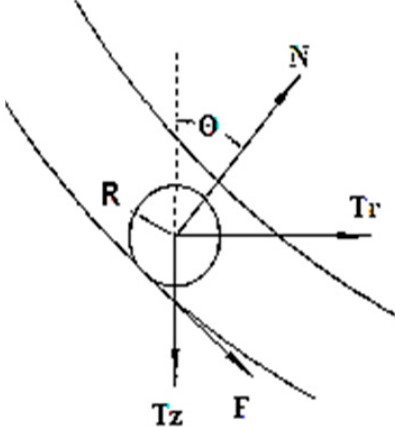

**Figure 7.** Simplified acting force on the main roller.

The static friction force F, determined by the force balance condition, is less than the maximum static friction force Fs [18], i.e.,

$$F < F_s, F_s = f_sN \tag{12}$$

That is,

$$f0 < fs \tag{13}$$

where f0 is the ratio of the friction force F to contact pressure N, and fs is the static friction coefficient [19]. The value of fs does not exhibit significant variation. Thus, in the design process, finding the value of F and N is critical in order to judge whether the condition in Equation (13) is satisfied.

Taking the main roller as a free body, the momentum moment equation of the center of mass is expressed as:

$$F \cdot R = Mg \tag{14}$$

Thus, we have

$$F = \frac{iJ\omega_1{}^2\psi(\theta)}{R^2}\left[\ddot{i}./i + \alpha/\omega_1{}^2 - \varphi(\theta)\dot{\theta}\right] \tag{15}$$

To obtain the value of N, Newton's Second Law needs to be applied to the acting force in the longitudinal direction of the bolt body. F and N are projected in the bolt motion direction and form the active push force

$$N_z = N\cos\theta - F\sin\theta \tag{16}$$

and the cross-direction force

$$N_\tau = N\sin\theta + F\cos\theta \tag{17}$$

The two forces are vertical to the motion direction. In addition to the friction force due to inertia forces, other forces and their spatial positions are illustrated in Figure 8. The symbol h represents the distance from the main roller mass center to the central line of the guide rail on the bolt body, b represents the length of the bolt guide rail, and c is the width of the bolt body. The quantities *x0* and *y0* represent the longitudinal and horizontal distance, respectively, from the main roller mass center to the bolt body mass center.

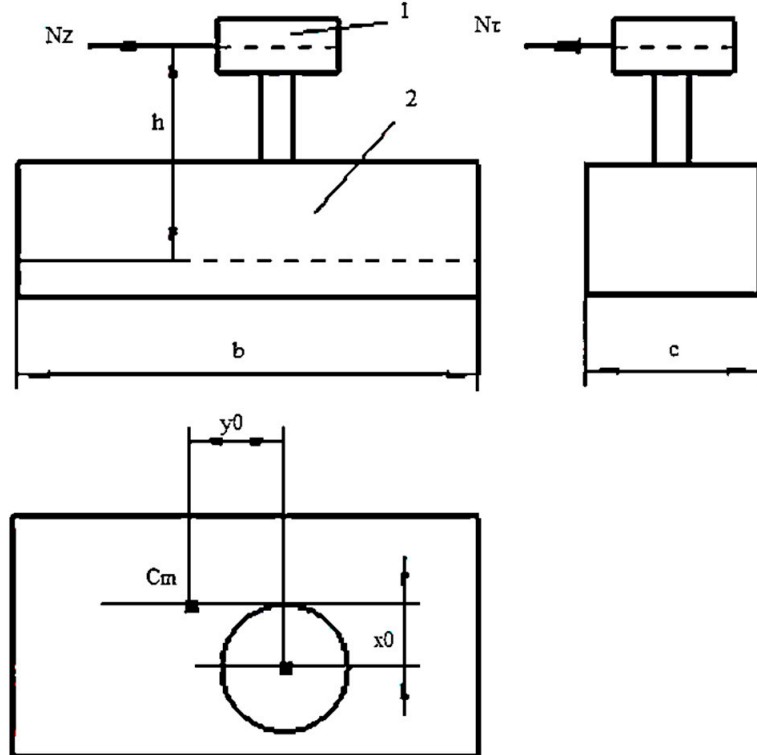

**Figure 8.** The active force of the bolt body. Note: 1—main roller, 2—bolt body.

The forces, including the driving force *F* and the friction force *Fs*, are projected in the longitudinal motion direction of the bolt body. Based on Newton's Second Law, the motion equation of the bolt body is as follows:

$$ma = N_z - 2f\frac{N_z h}{b} - 2f\frac{N_\tau h}{c} - 2f\frac{N_z x_0}{b} - 2f\frac{N_\tau y_0}{b} - fmr_1\omega_1{}^2 - fmr_1{}^2\alpha_1 - fN_\tau \quad (18)$$

$$N = \frac{1}{A\cos\theta - B\sin\theta} \times [F(A\sin\theta + B\cos\theta) + C] \quad (19)$$

$$A = 1 - 2f\frac{h}{b} - 2f\frac{x_0}{b} \quad (20)$$

$$B = 2f\frac{h}{c} + 2f\frac{y_0}{b} + f \quad (21)$$

$$C = ma + fmr_1\omega_1{}^2 + fmr_1{}^2\varepsilon_1 \quad (22)$$

In Equation (19), the value of the normal force *N* is obtained. According to Equations (13) and (14), it achieves:

$$N/F > 1/fs \quad (23)$$

According to Equations (18) and (24) can be derived from Equation (23).

$$\frac{f_s}{A\cos\theta - B\sin\theta}[(A\sin\theta + B\cos\theta) + \frac{CR}{J\varepsilon}] > 1 \quad (24)$$

Equation (24) presents the necessary condition for the roller rolling. For the designed cam–roller device, it can be used as a criterion to evaluate the design quality of the cam mechanism because the pressure angle $\theta$ is a function of the revolving angle $\phi$. Based on this information, designers can know whether or not the main roller is pure rolling on the corresponding angle $\phi$. Additionally, during the design process, the designer can adjust the size and shape of the cam curve to make the roller roll in a heavy-load region [7,20].

The necessary condition of the roller group in the pure rolling state is that every main roller is pure rolling [7,13]. The pure rolling criterion of the roller group is expressed as

$$\prod_{i=1}^{K} f_s \left( D + \frac{CR}{J\varepsilon_i(A\cos\theta_i - B\sin\theta_i)} \right) > 1. \quad (25)$$

where K is the number of set-A, set-B, and set-C, as shown in Equation (19). D is expressed as follows:

$$D = \frac{A\sin\theta_i + B\cos\theta_i}{A\sin\theta_i - B\sin\theta_i} \quad (26)$$

The main roller is a cylinder with a thickness of t, a density of $\rho$, an outer radius of R, and an inner radius of Rn. The rotary inertia J can be expressed as

$$J = \frac{1}{2}\rho\pi t(R^4 - R_n^4) \quad (27)$$

By $A\cos\theta - B\sin\theta > 0$, and substituting the expression of $\varepsilon$ in Equation (8), the expression of Equation (27) is taken into Equation (25). Equation (25) then takes the following form:

$$\prod_{i=1}^{K} fs \left( D + \frac{CR}{F_{J\varepsilon}{}^i(A\cos\theta_i - B\sin\theta_i)} \right) > 1. \quad (28)$$

where K is the number of follower rollers, and

$$F_{J\varepsilon}{}^i = \frac{i\rho\pi t(R^4 - R_n^4)\omega_1{}^2\psi(\theta_i)\lambda(i, \theta, \alpha, \omega_1)}{2R^2} \quad (29)$$

$$\lambda(i, \theta, \alpha, \omega_1) = \dot{i}./i + \alpha/\omega_1{}^2 - \varphi(\theta_i)\dot{\theta}_i \tag{30}$$

According to the expression of J in Equation (27), it can be seen that a decrease in the main roller's thickness t, an increase in the inner radius Rn, and a decrease in the outer radius R help to reduce the friction power consumption of the roller rolling. As shown in Equation (28), the roller group rolling criterion of a cam mechanism of the automaton is derived.

## 3. Systems Mechanism Kinematics Analysis

Without considering the influence of recoil motion, the device can be run during the launching process, as shown in Figure 1. The main roller associated with the bolt transfers the cam curve slot. The force acts on the push action surface in the acceleration period. The normal force becomes zero on the point with an acceleration of zero. Because of the clearance, the main roller runs to the other envelope plane of the cam curve and has a cross-impact. Additionally, as shown in Figure 1, during the firing stroke, the cam slot and the bolt group recoil are accelerated by the recoil spring. During the pushing motion, the bullet is pushed into the bolt body. In the process motion of the shell group, the shell is considered as the bolt body. In this case, when the motion acceleration of one cam roller device is greater than 0, the device is forced out of the roller surface and pushed across the envelope surface [7,20]. This affects not only the group motion but also the systemic resistance moment.

### 3.1. Analysis of Three States

Figure 9 illustrates the face-to-face contact process of the main roller and the cam curve slot. According to the motion equation in the direction of *F* and the inertial force *Fi*, some findings are obtained. The critical point A is pushing. The roller is rolling before it passes point A; when it runs across point A, the roller is in the pushing stage. Therefore, point A is the critical point where the state of a roller changes, and it is denoted as CPA. The face-to-face contact of the cam and the roller is on the rear-end surface of the cam curve slot and is vertical to the force *Fi*. The motion differential equation of the cam–roller device can be expressed as

$$dM_g{}^i = dF_i \times R \tag{31}$$

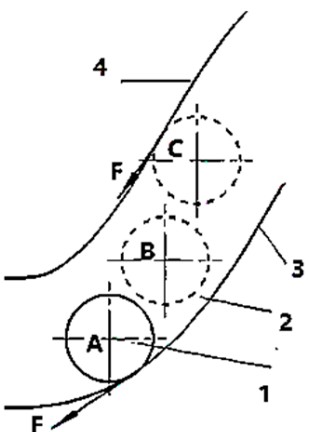

**Figure 9.** Three states of the main roller and cam curve slot. Note: 1—cam curve slot, 2—main roller, 3—rear-end surface, and 4—front-end surface. A—The critical point A, B—The critical point B, C—The critical point C.

When a roller is on point B, it is in the state of rolling. Before it runs across point B, the roller is in the pushing stage or the reverse pushing stage. Thus, point B is the beginning point of the rolling state, and it is denoted as CPB. Once a roller rotates through the CPB point, the roller runs clockwise, and the acting force F is the static friction force $F_f$ with a decreasing, rotating angle velocity. The face-to-face contact of the cam and the roller is on

the inner-end surface of the cam curve slot and is vertical to the force *F*. The differential equation is as follows:

$$dM_g^i = -dF_f^i \times r \tag{32}$$

The critical point C is in the reverse pushing state (CPC). Similarly, the C point is also critical, and it is denoted as CPC. It is rolling before a roller runs across point C. When it passes point C, the roller is pulling. Meanwhile, the friction force $F_f$ and the rotating angle velocity $\omega$ decrease. The face-to-face contact is on the front-end surface of the cam curve slot and is vertical to the force F. The differential equation can be expressed as

$$dM_g^i = -dF_i \times R \tag{33}$$

During a long period, the main roller has a variable rotating speed, and it could rotate clockwise or counterclockwise. If the contact time is short, the main roller keeps moving in the same direction. Additionally, the chassis moves forward, and the central roller moves out of contact. Then, it returns to CPA and enters the pushing contact state again. The duration is not only closely related to the motion of the recoil chassis but also directly associated with the design of the buffering [7,20].

Usually, the cam curve slot does not strictly follow the acceleration and deceleration derived from the contact surface wear of inverse transmission, and the cam curve slot on both sides experiences wear, for two reasons: On the one hand, the friction force is the driving force of the transmission agent. Therefore, the main roller brakes and then accelerates in reverse, accompanied by the wear of the cam curve slot in the A state to the C state. On the other hand, the main roller brake forces $N\tau^i$ and $Nz^i$ are not the pushing force on the bolt body.

*3.2. Several Bolt Groups Drive Transmissions with Recoil Clockwise and Counterclockwise*

With a clearance of the cam curve slot of the driving system of the chassis [11], the main roller enters the reverse state in advance and has a sudden recoil in the rolling state under the recoil caused by the launching load. As shown in Figure 10, a five-middle-caliber Gatling gun is taken as an example for illustrative purposes.

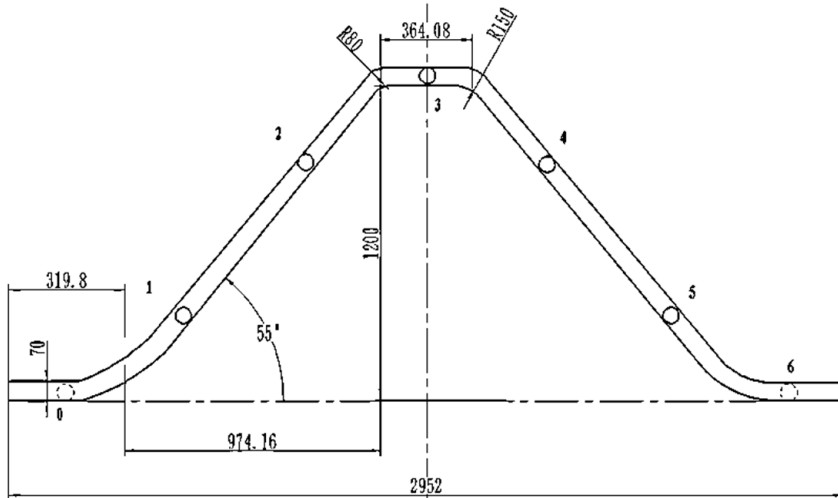

**Figure 10.** Influence of the relative motion of the roller on the device motion. Note: 1—positive acceleration critical point, 2—critical point of forwarding deceleration, 3—main roller in uniform speed stage (launch stroke), 4—reverse acceleration critical point, 5—reverse deceleration critical point, 6—(left) bomb position, (right) shell position.

Five sets of bolt bodies are arranged on the curve groove to achieve bomb loading, locking, stroke, unlocking, drawing shell, and throwing shell action according to the schematic diagram of the automaton. There are five rollers and bolt bodies on points 1, 2, 3,

4, and 5. The rollers and bolt bodies at the first and fifth points are pushed, corresponding to the acceleration and ejection acceleration stage. The rollers and bolt bodies at the second, third, and fourth points are inversely transmitted, corresponding to the pushing and ejecting deceleration stage. As shown in Figure 8, due to the clearance between the main roller and the cam curve slot, the buffer device fixed on the chassis is the releasing load in the direction of the arrow. After the advance to the inverse transmission state, the main roller with clearance runs on the first or fifth points. However, the main rollers on the second, third, and fourth points maintain the oppressive state. In one cycle, the system is in the instantaneous current state, where the two rollers on the third and fourth points are in positive drive, and the other three rollers on the first, second, and fifth points are in drive or drive from the inverse pushing state. This routine analysis does not involve the chassis of recoil with clearance or the several different cam curve slots.

The main roller does not run across the surface contact on the cam envelope during the recoil motion under high-frequency and high-speed vibration. Instead, it is located on the first and fifth points before reversing and recoiling. The main roller must be in the intermediate free state on the B position, as shown in Figure 9. The differential matrix equation of the roller group is as follows:

$$diag(dM_g{}^i) = -R_n \cdot dF_f^i, i = 1, 2, \cdots, K(K = 5) \tag{34}$$

After a period, the rollers are not in the same stage. It is difficult to obtain the differential matrix equation of the roller group. It is necessary to include the inertia moment of the five critical points, as shown in Figure 10. In addition, an external function is required to reflect the change in $dMg$. With an increase in the working cycles, the wear between the main roller and cam curve groove becomes increasingly serious. Then, the roller enters the motion distortion state, and finally, reaches a failed working state. Thus, it is necessary to diagnose the motion distortion of the roller–cam device of the Gatling gun automaton.

## 4. Research on the Motion Distortion Identification of Cam Mechanism

As one of the leading technologies of unsupervised machine learning, the fuzzy clustering algorithm can be adopted to conduct analyses, build models, establish the sample uncertainty genus, and objectively reflect the real world. Because of its significant theoretical and practical application value, this method has been effectively and widely applied to large-scale data analysis, data mining, vector quantifying, image segmentation, pattern recognition, etc. With further development, the fuzzy clustering algorithm has become a research hotspot [21–23]. The fuzzy C-mean clustering algorithm requires a relational membership weight for each cluster's data point (element). The sum of the membership weight of all the data points is 1. In essence, this method specifies the membership weight of each data point in each cluster manually and randomly. Then, it calculates the center of every group based on the membership weight and updates the membership weight matrix until the centroid does not change. The absolute value of all membership weights must be lower than the threshold set in advance.

This study recommends a new fault identification method, i.e., Variational Box Dimension Kernel Fuzzy Mean Clustering Algorithm (VK). The N-S flowchart of the VK fault identification of the cam mechanism is shown in Figure 11. The method consists of four steps. The first step performs module decomposition and feature extraction from original experimental data through variational mode decomposition [24] and fractal box-counting dimension (VMD-FBCD) [25].

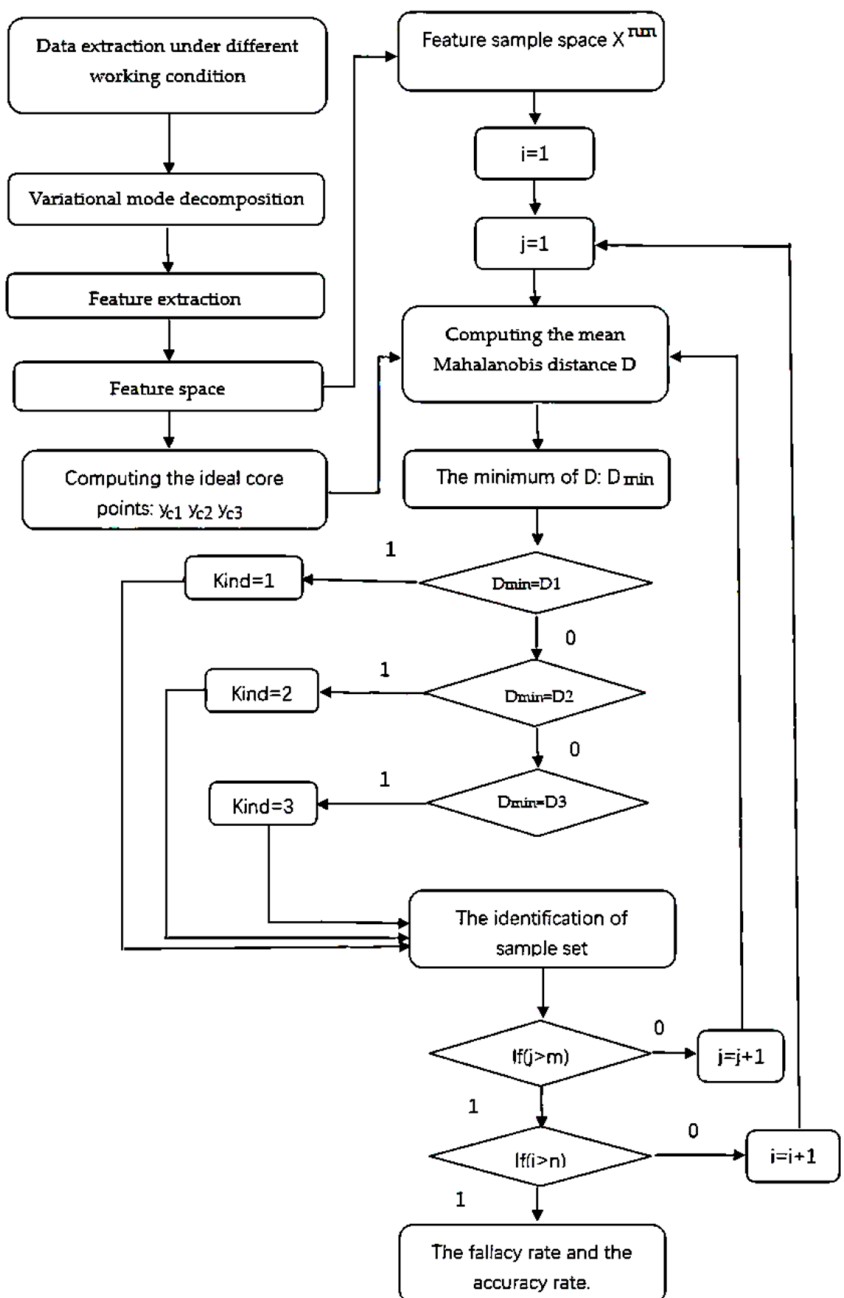

**Figure 11.** N-S flow chart.

The second step is searching for the ideal center of every working model by using the KMFC algorithm. The third step is building the feature samples space (FSS). The fourth step is identifying the faulty mode of every sample from the FSS by using the KMD-FC algorithm. The method is applied to the module decomposition and feature extraction of the experimental data pool of the cam–roller mechanism in a working state. The extraction results are the same as those in Equation (35).

$$X = [x_1, x_2, x_3, x_3, \ldots, x_m] \tag{35}$$

### 4.1. Motion Distortion Module Identification by KMD-FC Algorithm

KMD-FC is a type of novel fuzzy clustering algorithm with a kernelized Mahalanobis distance [26]. The method is suitable for failure mode recognition. The KMD-FC algorithm finds the best center by fuzzy clustering and calculates the minimum Mahalanobis distance

from the center to the specified system. Similarly, whether a sample is in a duty or fault state can be judged by calculating the distance from the sample data to the normal and the faulty operation centers. The smallest distance is applied to judge the working state of the sample.

*4.2. Modification of the Convergence Coefficient of the Kernel Fuzzy Clustering Function*

Suppose we have a sample space $X$, a vector space containing $n$ vectors. Vector $X_i$ contains m independent variables.

$$X = [X_1; X_2; X_3; \ldots; X_i; \ldots; X_n] \quad (i = 1, 2, 3, \cdots\cdots, n) \tag{36}$$

The average value of the samples is obtained as

$$\overline{X} = \frac{1}{n}\sum_{i=1}^{n} X_i. \tag{37}$$

A central point could be pointed, which is in the neighborhood of the o point. Here, the domain value is $\delta$. It is as follows:

$$X_C = \overline{X} + \delta \tag{38}$$

$$\delta = [\delta_1, \delta_2, \ldots, \delta_i, \ldots, \delta_m] \tag{39}$$

The distance between the sample space and the supposed ideal nuclear center $Pc$ point is denoted as $Dc$. The calculation formula of $Dc$ is as follows:

$$D_c = \sqrt{\sum_{i=1}^{n}\sum_{j=1}^{m} (X_i - \overline{X} - \delta)^2}. \tag{40}$$

Since the $Pc$ point is constantly changing, the change in $Dc$ is studied, supposing N is equal to 1. A one-turn iteration adopts the nuclear center point P1 from the o point.

When $N$ is equal to $t$, $t$ is a positive integer, and is considered as the $Pt$ point. The distance is

$$D_{c,t} = \sqrt{\sum_{i=1}^{n}\sum_{j=1}^{m} \left(X_i - \frac{1}{n}\sum_{i=1}^{n} x_{i,j} - \delta_{j,t}\right)^2}. \tag{41}$$

The approximation function $\Phi(X)$ is expressed as

$$\Phi(X) = D_{c,t+1} - \lambda D_{c,t} \tag{42}$$

Its numerator is

$$Y = \sum_{i=1}^{n}\sum_{j=1}^{m} \left(X_{ij} - \frac{1}{n}\sum_{i=1}^{n} X_{ij} - \delta_{i,t+1}\right)^2 - \lambda^2 \sum_{i=1}^{n}\sum_{j=1}^{m} \left(X_{ij} - \frac{1}{n}\sum_{i=1}^{n} X_{ij} - \delta_{i,t}\right)^2 \tag{43}$$

When y tends to zero, the ideal center point appears. Supposing the minimum value of y is 0.000001, the numerator $Y$ tends towards zero.

The partial derivative value of $Y$ is zero $\partial Y/\partial t = 0, \partial Y/\partial \lambda = 0$. Its deformation is

$$2\sum_{i=1}^{n}\sum_{j=1}^{m} \left(x_{ij} - \frac{1}{n}\sum_{i=1}^{n} x_{ij} - \delta_{i,t+1}\right)^2 \cdot \left(-\frac{d\delta_{i,t+1}}{dt}\right) + 2\lambda^2 \sum_{j=1}^{m}\sum_{i=1}^{n} \left(x_{ij} - \frac{1}{n}\sum_{i=1}^{n} x_{ij} - \delta_{i,t}\right)^2 \cdot \left(\frac{d\delta_{i,t}}{dt}\right) = 0 \tag{44}$$

$$-2\lambda \sum_{j=1}^{m}\sum_{i=1}^{n} \left(x_{ij} - \frac{1}{n}\sum_{i=1}^{n} x_{ij} - \delta_{i,t}\right)^2 = 0 \tag{45}$$

Denote $\delta$ as the linear change in t. The rate of change can be expressed as

$$\delta = [w_1, w_2, \ldots, w_j, \ldots \ldots w_m] \cdot v \cdot t_0 + \delta_o \tag{46}$$

Furthermore, it is called the approximation parameter or approximation factor. The quantity $\delta 0$ is the initial value of the distance between the O point and *Pc*. It equals 0 or the norm maximum of the boundary value $(Xi - \overline{X})$ minus the minimum value. $w$ is the weight factor matrix

$$w = [w_1, w_2, \ldots, w_j] \tag{47}$$

The boundary value $(Xi - \overline{X})$ is obtained by singular value decomposition. *S* represents m eigenvalues of the singular value decomposition. The diagonal array element of *S* is shown as follows:

$$diag(S) = [s_1, s_2, s_3, \cdots, s_j, \cdots, s_m] \tag{48}$$

Let the weight factor be

$$w_j = s_j / \sum_{j=1}^{m} s_j. \tag{49}$$

when these unknown parameters are substituted into Equation (45), the equation can be converted into a monadic quadratic equation with one unknown parameter, $v$. The value of $v$ can be obtained based on Equation (46). Finally, the position of the ideal cluster nuclear center is calculated, and high-precision results can be obtained with a high-performance computer. Only a micro-asymptotic ascent model can be used for the Mahalanobis distance, so a new path must be considered.

Suppose the initial cluster nuclear center is at the o point. The distance between the sample space and the o point is as follows:

$$Do = \sqrt{\sum_{i=1}^{n} \sum_{j=1}^{m} (X_i - X_C)^2} \tag{50}$$

Denote an independent variable $x_i \in X^{n \times m}$; the distance between xi and its nuclear center $x_{cj}$ is as follows:

$$y = \left[ \sum_{i=1}^{n} (x_i - x_{cj})^2 \right]^{1/2} \tag{51}$$

Let $z$ be equal to the square of $y$. It is presented as follows:

$$z = \sum_{i=1}^{n} (x_i - x_{cj})^2 = \sum_{i=1}^{n} x_i^2 - 2x_{cj} \cdot \sum_{i=1}^{n} x_i + n \cdot x_{cj}^2 \tag{52}$$

Let

$$c_1 = \sum_{i=1}^{n} x_i^2, c_2 = \sum_{i=1}^{n} x_i, c_3 = n \cdot x_{cj}^2 \tag{53}$$

where $c_1$ and $c_2$ are the square-sum and sum of the *j*-th independent variable, and $c_3$ is the number of samples:

$$c_1 x_{cj}^2 - 2c_2 x_{cj} + c_3 = 0 \tag{54}$$

When $z$ tends to 0, Equation (52) becomes a unary quadratic equation, and the root $x_{cj}$ is solved based on Equation (54). Meanwhile, the nuclear center of the specific system in the working state is formed. It is the center of the number-j independent variable $x_c$. A similar approach can obtain the accurate position of the ideal nuclear centers of the m-1 independent variables.

## 5. Experiment Verification

### 5.1. Case Analysis

A dynamic model of the main roller and cam curve slot is built. The model consists of the fixed cam curve slot, the rotary device, and the bolt group. The recoil and the clearance are also needed. Before the recoil process of the load-transmitting chassis of the automaton is simulated, it is supposed that there is one curve slot on the cam, a fixed connection between the cam and the chassis, and a movable pair between the chassis and the ground.

As shown in Figure 12, the recoil displacement of the chassis changes from −5 mm to 10 mm, for a medium-caliber Gatling gun shooting at 1000 rpm.

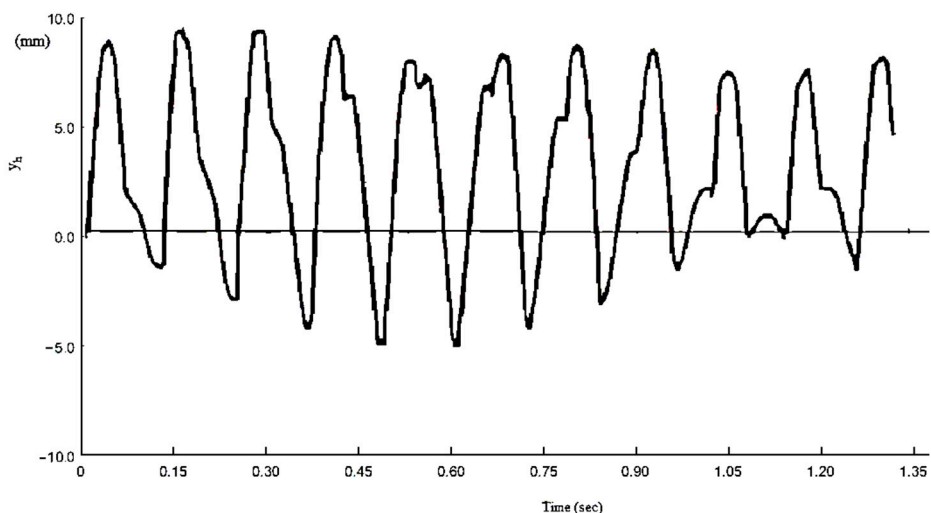

**Figure 12.** The recoil displacement curve, showing change in the Gatling gun vs. time.

Firstly, there is no clearance and recoil between the main roller and the cam curve slot. The analysis result is similar to that shown in Figure 13, and the roller displacement is also similar to the bolt body motion shown in Figure 3. In Figure 14, the analog result of the motion change in the main roller is presented. When there is no gap, the dynamic characteristics and the change in the contact force of the roller in the receipt are not considered, as shown in the following two figures. It can be seen from the figure that the angular speed, contact force, and roller displacement show periodic changes.

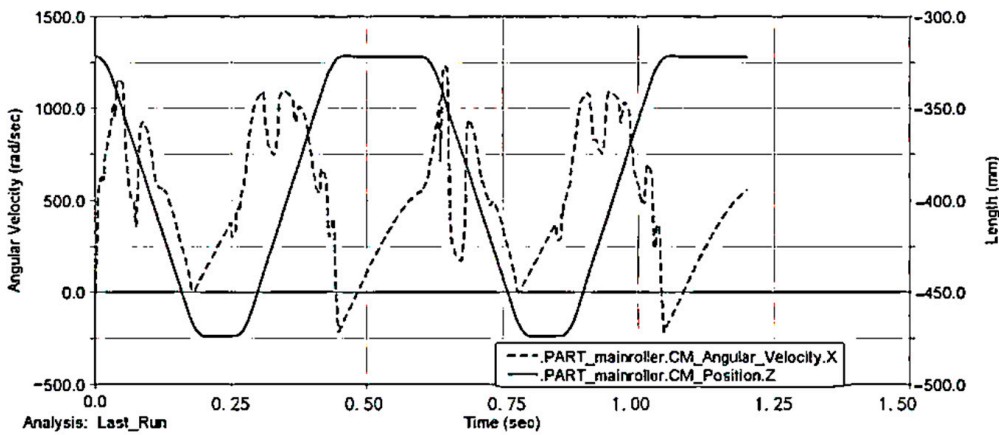

**Figure 13.** The motion change curve of the main roller without the consideration of recoil and gap.

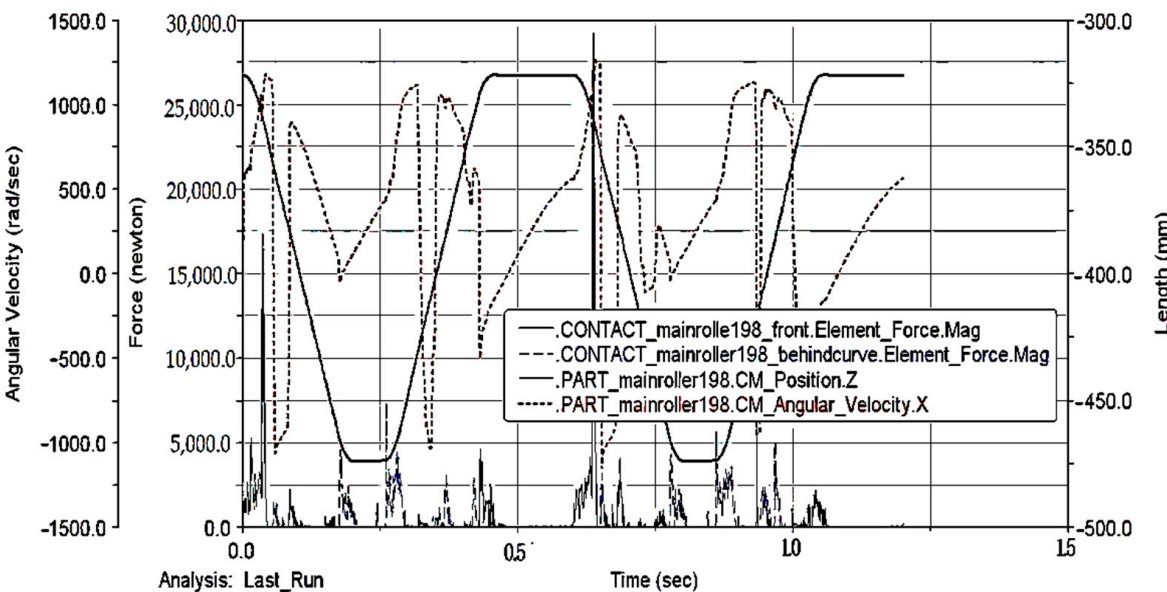

**Figure 14.** The motion change curve in the main roller with the consideration of gap and no recoil.

The dynamic characteristics of the main roller of the gap secondary cam mechanism are shown in Figures 13 and 14: the red dashed line represents the angular speed of the main roller, the solid black line represents the position of the main roller, the blue dotted line represents the change in the contact force of the main roller and the rear end of the curve groove, and the black dotted line represents the contact force of the change in the front end of the main roller. It can be seen that, because of the existence of the gap, the main roller and the two ends occur. After the firing rate is stabilized, the automaton gradually tends towards the periodic movement. During an average cycle, the main roller meets the front end once, and the contact force of the main roller and the rear-end surface changes periodically. The main roller cannot meet the conditions required for pure rolling, so only translation motion can occur, and not rolling; the contact surface is a finite domain inside contact, and sliding friction occurs.

Figure 15 illustrates the contact force variation of the front end and rear end of the cam slot curve with time t. The maximum value of the contact force exceeds 5000 N. The dotted line indicates the angular velocity of the main roller. The black line indicates the mass center of the main roller along the Z-axis between 475 mm and −375 mm.

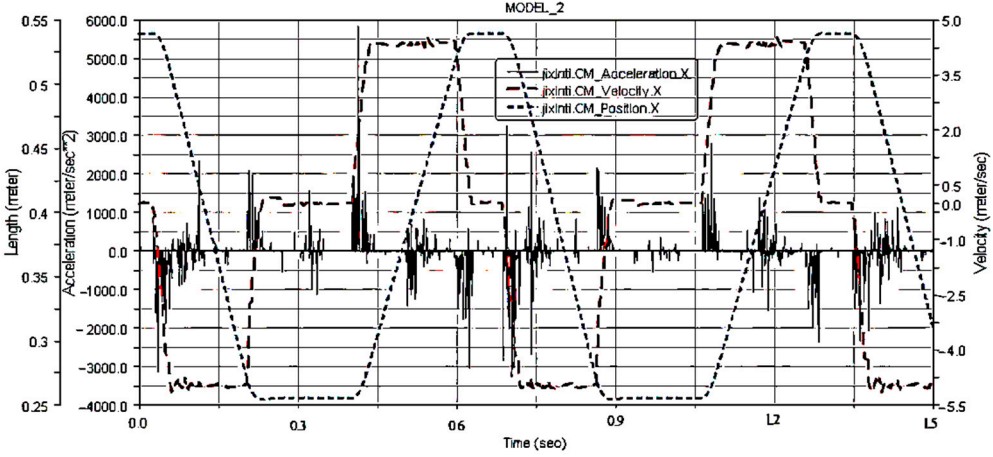

**Figure 15.** The change in the contact force vs. time without clearance and recoil.

The contact force between the main roller and the fixed cam curve slot is shown in Figure 15. Two dotted lines indicate the position and velocity of the mass center of the main roller, respectively. The black line indicates the variation in the acceleration of the mass center of the main roller with time t under no recoil. The maximum acceleration value is below 5000 m/s$^2$ except for the point at 0.406 s, and the motion period is 0.605 s.

With the clearance and under no recoil, the main roller motion is illustrated in Figure 16. The movement of the Gatling bolt body is illustrated in Figure 17. The time is extended to 1.5 s to observe the change in bolt acceleration. The clearance influences the system transmission in Figure 18.

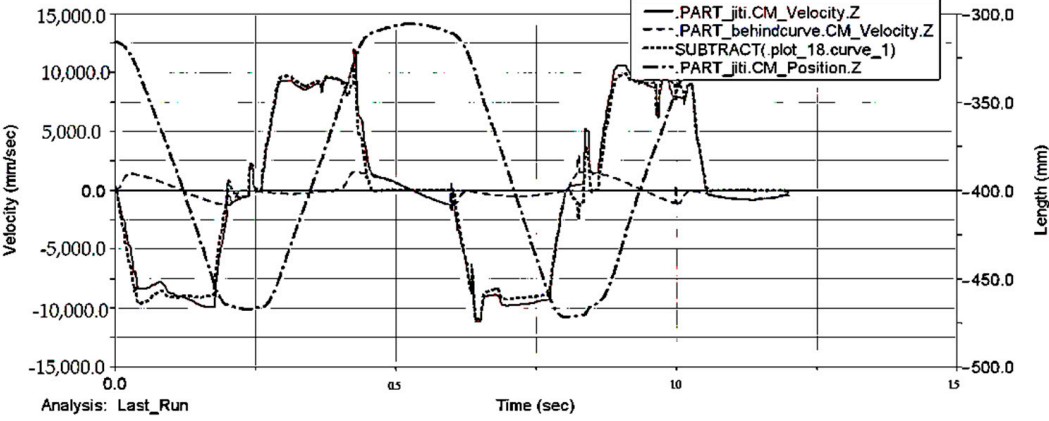

**Figure 16.** The bolt body motion of clearance and recoil.

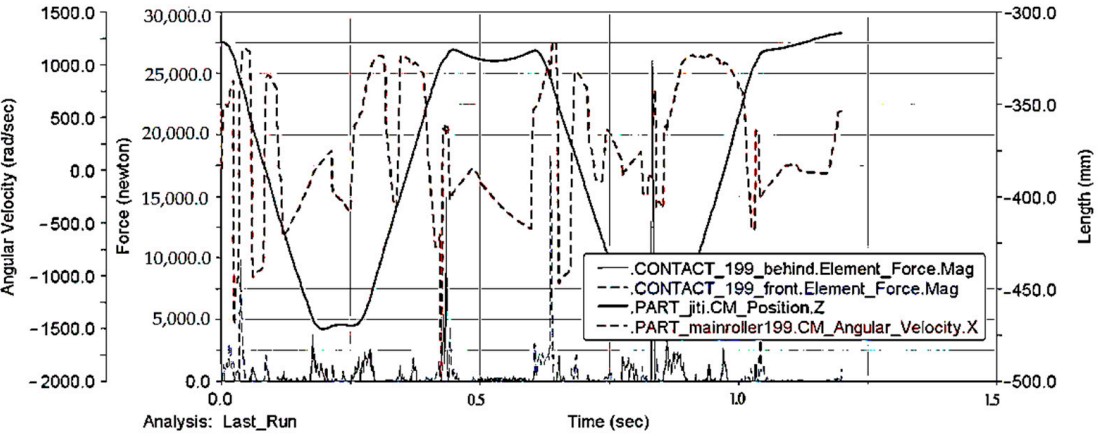

**Figure 17.** The bolt body motion without recoil and with a 0.1 mm gap.

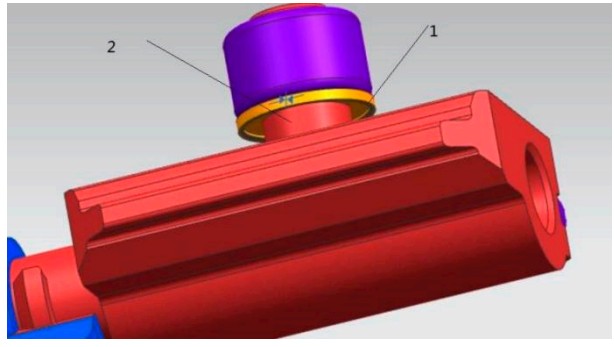

**Figure 18.** The bolt motion analysis with clearance and recoil. Note: 1—bolt body, 2—the rolling axis.

### 5.2. Test Method and Basic Structure

Considering the particularity of the Gatling gun, the storage testing technology is suitable for the acceleration testing experiment. The more comprehensive the testing range, the larger the layout cub-age of the test instrument, the higher the volume of the test instrument, and the larger the required space for instrument assembly.

However, because the structure of the actual head is optimized, the volume automaton is strictly restricted. For the same reason, it is also difficult to measure the bolt's acceleration, as shown in Figure 19.

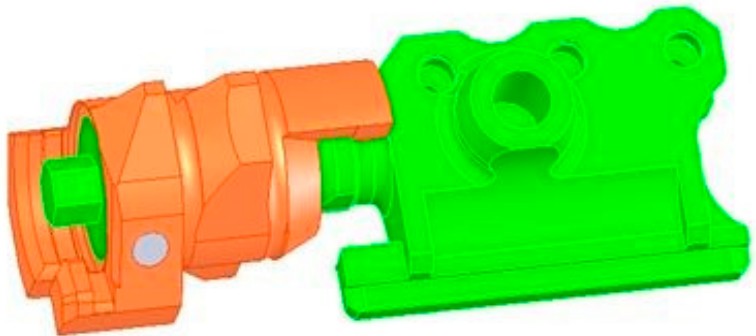

**Figure 19.** The recoil roller simulation with 0.1 mm clearance.

Thus, designing a storage test device with a smaller cub-age is necessary. Additionally, the acceleration sensor must be small in order to be installed in the improved core head accurately and easily. Otherwise, it is difficult to conduct the experiment when the main roller axes rotate around the axis of the gyrator, as shown in Figure 20.

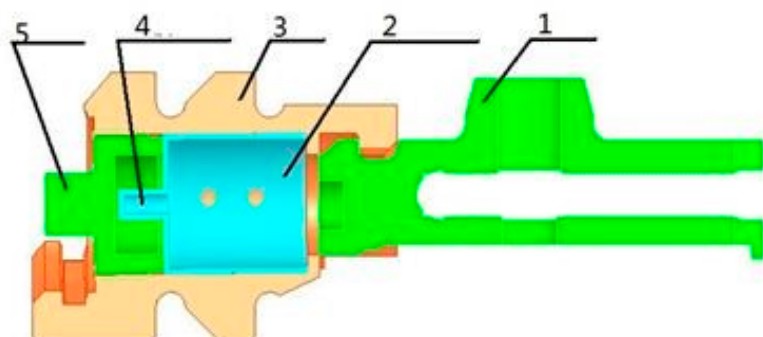

**Figure 20.** Test device structure. Note: 1—bolt body, 2—storage test device, 3—bolt head, 4—acceleration sensor, and 5—fixed lid.

The choice of acceleration sensor is made according to the parameters as follows: Based on theoretical analysis, the firing acceleration is 5000 $m/s^2$. The range of the acceleration sensor is 20,000 $m/s^2$. An internal trigger is selected as the testing trigger mode. When the maximum acceleration is 100 $m/s^2$, according to the requirements of testing time, the designed storage capacity needs a test time of 2 s. In addition, the testing time from the start point to the stable point of the firing rate needs to be known.

The local expansion diagrams of 1 and 2 of Figure 21 are consistent with Equations (31)–(33). Under the sampling frequency of 200 kHz, the acceleration of the bolt is shown below in Figure 22. The experimental result is shown in Figure 23a,b which is a local expansion diagram of 1 and 2 of Figure 21.

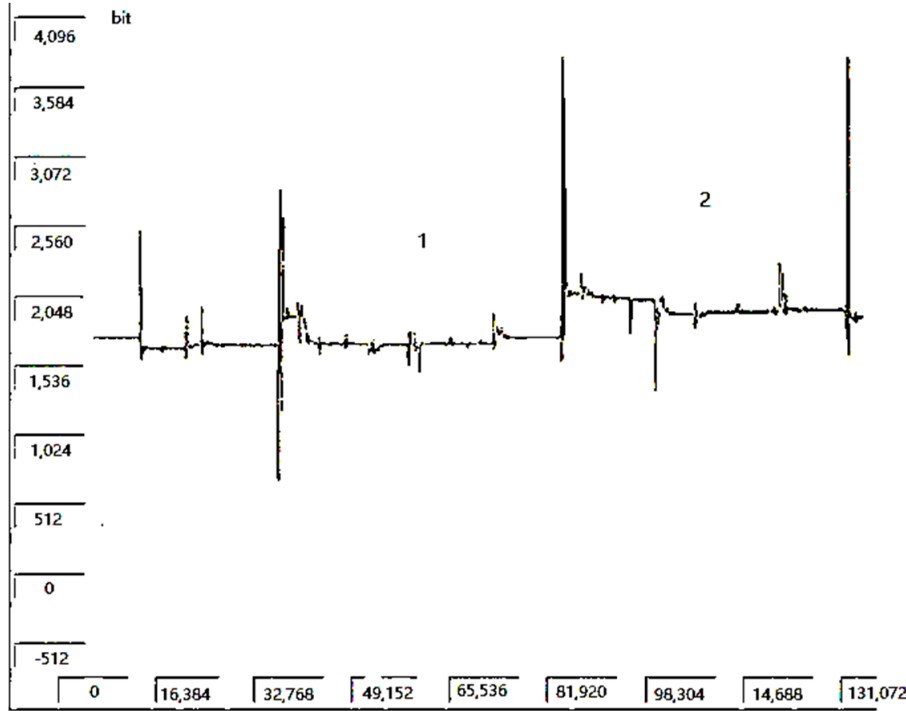

**Figure 21.** The main roller rotation experiment. Note: 1—graduated arc, and 2—main roller axis.

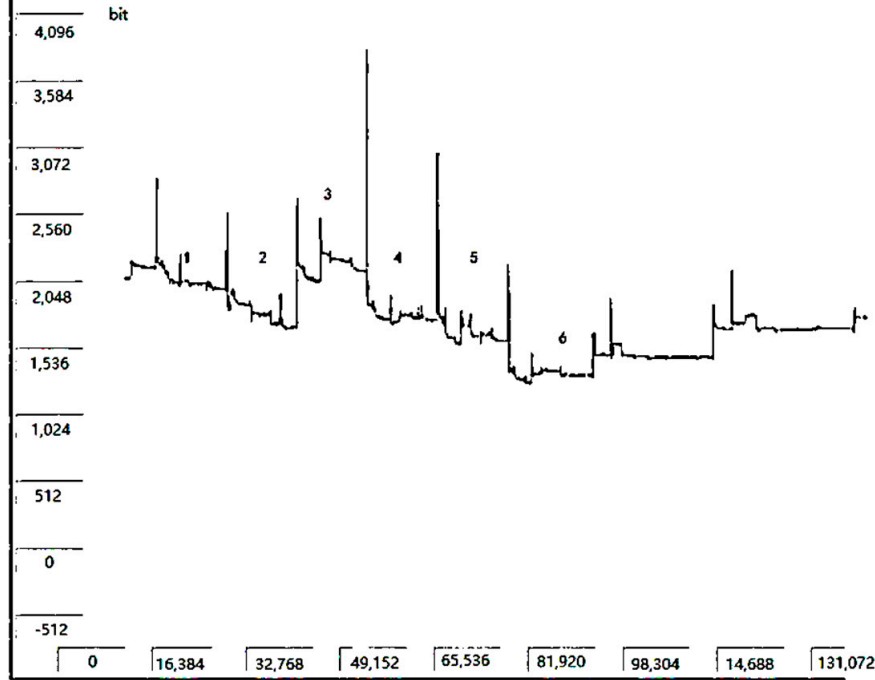

**Figure 22.** The Gatling bolt acceleration under the sampling frequency of 200 kHz. Note: 1—graduated arc, 2—main roller axis, 3—working part axis, 4—mechanism carriage axis, 5—graduated arc, 6—box mechanism axis.

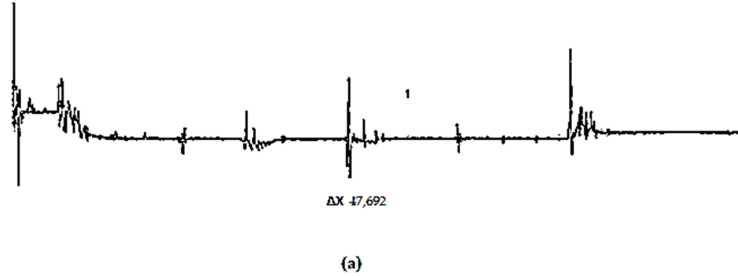

ΔX 47,692

**(a)**

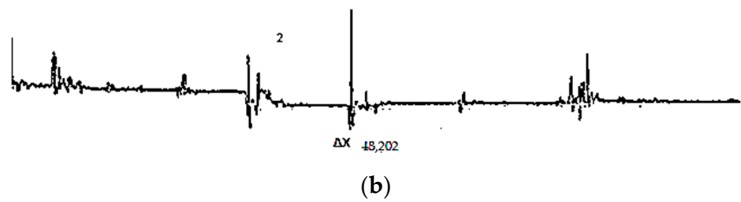

ΔX 48,202

**(b)**

**Figure 23.** Comparison of periods of acceleration in a test. Note: 1—graduated arc, and 2—main roller axis. Note: (**a**) the acceleration of the main roller in the graduated arc, (**b**) the acceleration of the main roller in the pushing stage.

The operation can be conducted with zero division and some noise removal. The acceleration can reach 80%, as shown in Figures 19–24. The result and method are verified to be correct. These results support that shown in Figure 21. The contact force of the main roller acted on the rear-end and front-end contact surface, as shown in Figure 12, and is considered the forward force. During the middle working cycles, it is verified that the five rollers cannot run in the same pushing or pulling state together. It is also shown that one or two rollers are in no rolling state at some points in the experiment. From Figures 22–26, the main frequency is 16,384 Hz, and the base frequency is 512 Hz. The processed picture is shown in Figure 26. The computational results are consistent with those shown in Figures 15–17, in which the image curve is remarkably similar.

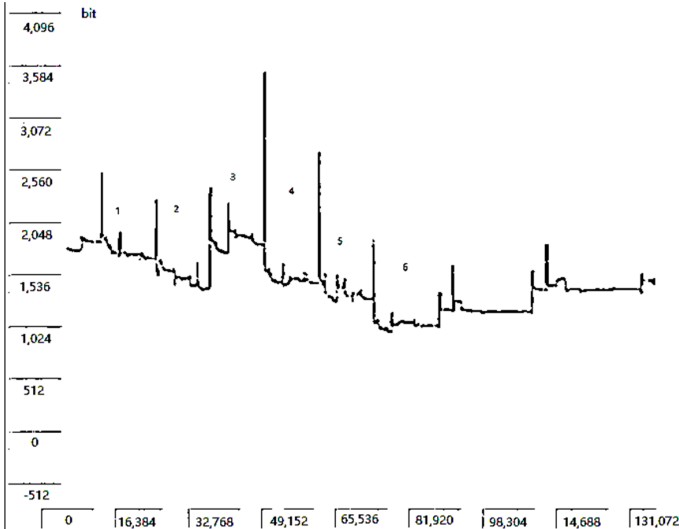

**Figure 24.** The Gatling bolt acceleration curve at the sampling frequency of 50 kHz. Note: 1—graduated arc, 2—main roller axis, 3—working part axis, 4—mechanism carriage axis, 5—graduated arc, 6—box mechanism axis.

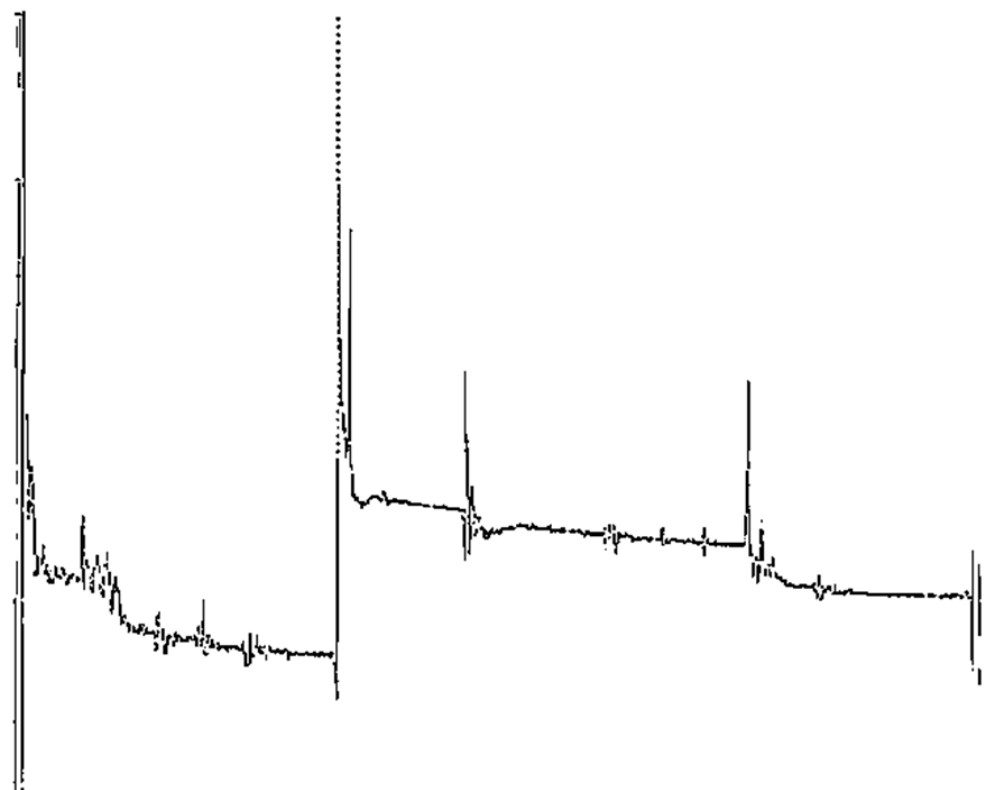

**Figure 25.** The amplified longitude acceleration of Figure 19.

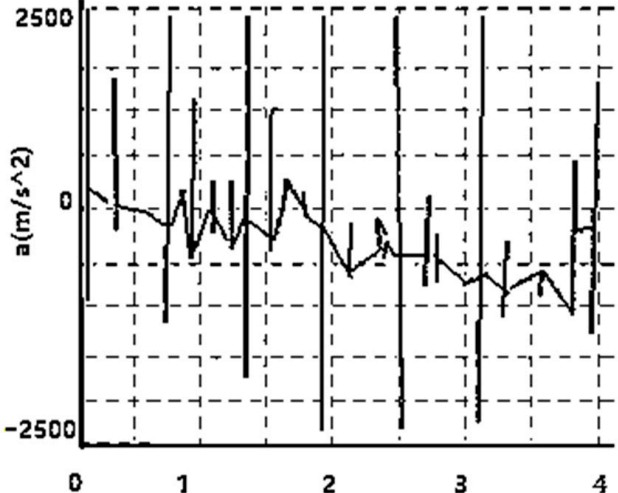

**Figure 26.** The experimental acceleration curve of the bolt body.

The movement trend of rollers of different sizes has a roughly similar trend in the cam curve groove, which shows a trend of periodic change. However, at the same position, the interface with a large primary roller radius corresponds to a large positive pressure. The Figures 14, 15 and 17 are R = R0, R = R0–0.05, and R = R0–0.1, respectively; in three cases, the changing trend of the contact force is roughly the same, but at the same position, the positive pressure of the contact surface of the interface is slightly larger, as is the side pressure after the collision with the cam curve groove.

As the number of working cycles increases, the wear between the main roller and the cam curve groove becomes serious. The broader their clearance, the higher the systemic energy consumption. Meanwhile, the more serious the wear of the guide groove and the

guide rail, the worse their motion consistency, and the easier the motion interference and cartridge jamming in the experiments. The face-to-face contact wear of both the roller-groove and the rail slot is so severe that the Gatling gun automaton gradually decelerates to zero and finally come to a halt, causing cartridge jamming.

It can be seen from Figure 27 that the ideal cores of the main roller of the three different working states can be achieved by using dmkf2 for the second modification of the convergence coefficient of the kernel fuzzy clustering function.

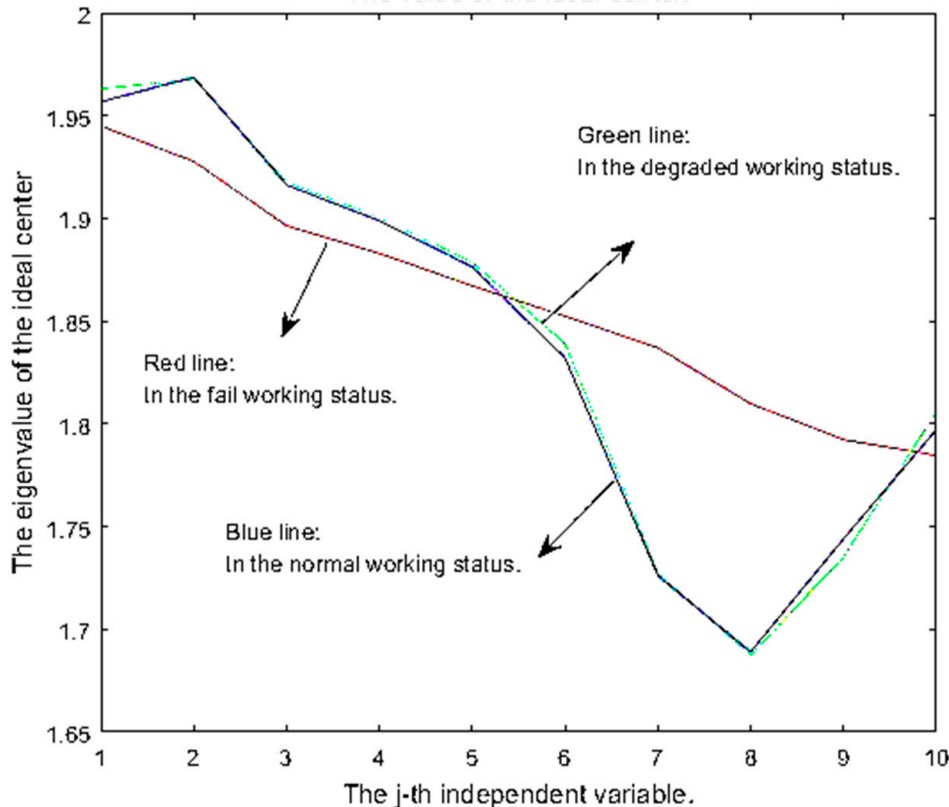

**Figure 27.** Three ideal systemic centers in the three working states of normal, deteriorated, and failure are obtained by dmkf2.

According to Table 1, there are 2810 samples in total, including 1560 samples in the normal working state, 870 samples in the degraded state, and 380 samples in the failure state. All the middle positions of the result matrixes A11, A12, B11, B12, C11, and C12 are zeroes. Using the kernel fuzzy clustering function for the first modification of the convergence coefficient, the pattern recognition accuracy of the 1560 duty samples is 42.4%, and the misjudgment rate is 57.6%.

**Table 1.** Comparison of the fault diagnosis results utilizing dmkf1 and dmkf2.

| State | Num. | dmkf1 | dmkf2 |
|---|---|---|---|
| Normal | 1560 | A11 | A12 |
| Degraded | 870 | B11 | B12 |
| Failure | 380 | C11 | C12 |

Note: A11—[42.4, 0, 57.6], A12—[98.59, 0, 1.41], B11—[36.7, 0, 63.3], B12—[99.77, 0, 0.23], C11—[44.5, 0, 55.5], and C12—[2.89, 0, 97.11].

A significant difference in using the kernel fuzzy clustering function for the second modification of the convergence coefficient can be observed. The misjudgment rate is 1.41%, and the accuracy is above 98.59%. A similar case occurs for the 870 degraded samples and 380 failure samples. By applying dmkf1, 36.7% of the former samples are recognized as

being in the normal state, leading to a misjudgment rate of 63.3%. However, the result is 99.77% and 0.23% by the method of dmkf2. Of the 380 samples in the failure state, 44.5 % are recognized as being in the normal state, leading to a misjudgment rate of 55.5%.

On the contrary, the result is 2.89 % and 97.11% when employing the dmfk2. The above experimental results indicate that the degraded sample data can be identified neither by dmkf1 nor dmkf2. They are either in the normal state or failure state, as determined by employing dmkf1 and dmkf2.

The total computation time is 43.4541 s, where 34.7969 s is for dmkf2 and 8.6563 s for fault identification. Approximately 80% of the time is used to seek the ideal core, and the module identification of 2810 samples of the Gatling gun automaton accounts for 19.92% of the time. Once the systemic centers are discovered, the recognition speed is approximately 0.003 s for each point of sample data, which is twenty times faster than the firing rate of the Gatling gun in the experiment.

## 6. Conclusions and Expectation

This paper studies the gap between the main roller and the curve groove; the larger the gap, the stronger the front and rear movement of the main wheel. At the same time, the greater the shear force of the main roller shaft, the more likely the shaft is to break. In contrast, the smaller the gap, the less energy of the movement. The lower the sheer force of the axis is, the better the protection of the axis is. However, in the case of too small clearance, the primary roller cannot rotate normally, and the higher performance of the lubricating oil also has been needed. This paper not only considers the gap between the roller and the curve groove but also considers the direction of motion. Therefore, based on the requirement for pure rolling in the roller group, we can rapidly predict when the rolling wheel sticks to the front and rear wall and when it is in the middle position. It provides favorable conditions for studying the wear of the cam curve groove of the automaton.

The failure cam mechanism is researched given the kinematic and kinetic views of the multi-body system. Meanwhile, clearance and recoil are considered in deriving the pure rolling formula of the roller group. This paper focuses on the wear of the curved groove of the cam and the frequent braking and rapid reversal of the main roller caused by recoil. The VK algorithm proposed in this paper enables fast and accurate model identification. Based on dmkf2, which can effectively distinguish between the motion distortion state and the normal working state of the multi-roller cam mechanism, the present results provide a reference for an insight into the wear mechanism of the complex mechanical system.

The data transmission through online testing and monitoring systems can be achieved in future work. Not only can it obtain the related information on both the gap and the movement characteristics of the mechanism in a timely manner, it can rapidly judge the current working conditions and quickly determine whether the roller or cam curve groove needs to be replaced. In future research, the degeneration of rollers with different strength materials can be studied in different lubrication environments to quickly replace the roller wheel, reduce the wear of the cam curve groove, save cost, and make new contributions to the development of the Gatling gun.

**Author Contributions:** Conceptualization, methodology, software, and formal analysis, writing—review and editing and communication, X.C.; Project administration and supervision, funding acquisition, and resources, H.P.; Validation, modeling, investigation and resources, J.X.; Experiment preparation, data curation and writing—original draft preparation, T.W. All authors have read and agreed to the published version of the manuscript.

**Funding:** The author gratefully acknowledges the support from the following foundations: National Natural Science Foundation of China (51175480 and 51675491), and the Opening Research Funds for the advanced manufacturing Key Laboratory of Shan-xi of China (XJZZ201701).

**Institutional Review Board Statement:** Not applicable.

**Conflicts of Interest:** The authors declare no conflict of interest.

## Nomenclature

| | |
|---|---|
| $\alpha$ | Angular acceleration of the revolving body. |
| $\Psi(\theta)$ | A function of $\theta$, $\csc(\theta)$. |
| $\delta$ | Neighborhood domain value of the o point. |
| $\delta 0$ | Initial value of the distance between the O point and Pc. |
| $\varepsilon$ | Angular acceleration of the main roller. |
| $\phi$ | Angle of revolution of the revolving body. |
| $\varphi(\theta)$ | A function of $\theta$, $\cot(\theta)$. |
| $\Phi(X)$ | Approximation function of X. |
| $\rho$ | Density of the main roller. |
| $\rho 0$ | Friction circle radius. |
| $\theta$ | Pressure angle matrix. |
| $\theta i$ | Motion pressure angle of the roller. |
| $\omega$ | Roller group angular velocity matrix. |
| $\omega_1$ | Angular velocity of the revolving body. |
| $Fi$ | Inertial force. |
| $Fs$ | Static friction force. |
| $fs$ | Static friction coefficient. |
| $J$ | Rotary inertia. |
| $K$ | Number of rollers. |
| $Mg$ | Inertia moment. |
| $N$ | Vertical contact force. |
| $n$ | Firings or the number of cycles. |
| $Num$ | Abbreviation of numbers of samples. |
| $Pc$ | Ideal nuclear center Pc. |
| $R$ | Main roller radius. |
| $Rn$ | Radius of the pin roll. |
| $S$ | Eigenvalues of singular value decomposition. |
| $t$ | Thickness of a cylinder. |
| $V$ | Mass transitional velocity matrix. |
| $Vz$ | Longitudinal velocity vector of the bolt. |
| $v$ | Rate of change in the weight factor. |
| $w$ | Weight factor matrix. |
| $X$ | Feature vector group. |
| $Y$ | Numerator of the transformation of the function $\Phi(X)$. |

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
