# Peer review of "Study of Fault Identification of Clearance in Cam Mechanism"

_applsci, doi:10.3390/app12157420_

Round 1
Reviewer 1 Report
I recommend this paper to be published in this Journal pending major revision:
(1) All the figures are blurry and need to be clearer.
(2) How the author describe the impact model between the fixed cam curve groove, roller follower, and the bolt at high speeds in the presence of clearance.
(3) What is the effect of pure rolling and sliding on the dynamic characteristic of the roller follower in the presence of clearance at high speed?
(4) The author claimed that if there will be no clearance, the dynamic motion of the system should be periodic. More explanation is needed. In the presence of clearance the system will move with non-periodic motion and chaos.
(5) In figures 17-21 I think the author studied the effect of acceleration against frequency using Fast Fourier Transform analysis but he did not discussed anything about the dominant frequency and fundamental frequency.
(6) The author claimed that he used different sizes for the roller follower. Could you please explain what is the effect of this different rollers’ sizes on the positive pressure?
(7) In the reference section there are two numbers which affects the citation. Which number did the author take? Please check the title of reference number (4) especially the word (Global) it should be (Globoidal).
Author Response
Applied Sciences Editorial Office<applsci@mdpi.com>
Article: Fault Mechanism and Identification Study for Cam Mechanism of Clearance
Manuscript ID: Applsci-1761071
NUC., Taiyuan, Shanxi, China
19th June 2022
Dear reviewer,
It is the first appreciation of the advice of the respected anonymous reviewers. According to your seven pieces of advice, the article manuscript has been revised. The following is in detail.
The change includes six sections. First is the background introduction second. It is in the first part. Then is the reference documentation; we omitted the references before 2006 and reminded those for 15 years, and renewed the reference section. The third part is an additional detail. They are on pages 1, 5, 16, 17, 22, 23, 25, and 26. Detailed locations are shown on the diagram introduction of the revised part22022.06.19 (DIRP, in the form of pdf). The revised part is dotted with red letters. The blue letters are the reviewer's questions (Q) and my answers (A). The black letters represented the main body of the article. In addition, all the changed marks have remained in the submitted doc fil. 90% of all figures have been renewed for the time limit. In addition, we also resort to the MDPI English language service.
Next, this is an answer to the first reviewer. Please refer to the DIRP.
Q1: All the figures are blurry and need to be more explicit.
A: Due to the limited time, only 90 percent of the pictures have been updated, and if necessary, continue to update them later.
Q2: How the author describes the impact model between the fixed cam curve groove, roller follower, and the bolt at high speeds in the presence of clearance.
A: A cam roller mechanism model with a gap is given with the necessary text description. (DIRP, page 5)
Q3: What is the effect of pure rolling and sliding on the dynamic characteristic of the roller follower in the presence of clearance at high speed?
A: Because of the gap, the collision between the main roller and the two end surfaces often occurs. During an average cycle, the main roller meets the front end once, and the contact force of the main roller and the rear end surface changes periodically. The main roller cannot meet the conditions for pure rolling, so the rolling cannot occur, and translational motion can only do. The contact surface is a finite domain inside contact with sliding friction. (DIRP, page 17)
Q4: The author claimed that if there is no clearance, the dynamic motion of the system should be periodic. More explanation is needed. In the presence of clearance, the system will move with non-periodic motion and chaos.
A: Looking closely at Figs 14 and 15 to know this point. It can also be obtained from the law of numerical change. (DIRP, page 16)
Q5: In figures 17-21, I think the author studied the effect of acceleration against frequency using Fast Fourier Transform analysis, but he did not discuss the dominant frequency and fundamental frequency.
A: the dominant frequency is 16384 Hz, and the fundamental frequency is 512 Hz. The related signal processing is not the critical point of the article. We chose to simple introduction.
Q6: The author claimed that he used different sizes for the roller follower. Could you please explain the effect of these different rollers' sizes on the positive pressure?
A: In the following three situations of R=R0, R=R0-0.05, and R=R0-0.1, the changing trend of the contact force is roughly the same. However, at the same position, the positive pressure of the roller with a large radius corresponds to a slightly more tremendous positive pressure of the contact surface of the interface. The side pressure after colliding with the cam curve groove is a little smaller.
Q7: In the reference section, two numbers affect the citation. Please check the reference number (4) title, especially the word (Global). It should be (Globoidal). Which number did the author take?
A: Revised all the related references and contents.
Finally, thanks a lot for all the advice again. If any questions, please contact me at 20080005@nuc.edu.cn.
Best wishes to you.
Yours sincerely,
Xuefang Chang

Reviewer 2 Report
Though the research sounds interesting, there are several issues that must be addressed. Among others, it is poorly presented and the novelty is not justified.
Specific comments:
1. The title is vague. Consider to modify as "Study on Identification of the Fault Mechanism of Cam Mechanism Clearance".
2. The abstract requires more concretizing.
3. Introduction is too brief. Sufficient literature is not explored.
4. Line 47 - 50: Move the description of the numbered parts of the figure either beside the figure or just below the caption text. Apply this to other figures of similar description.
5. Equal sign is missing in Equation (7).
6: The heading title of Section 3 is unclear "Systems mechanism kinematics analysis".
7. Part of the text in flowchart in Fig 9 is diffuse. Please clear the red underlines
8. Figures 11 - 14 are partly unclear legend texts are invisible (unreadable).
9. Figures 17 - 22 are not sufficiently discussed.
10. Many sentences are difficult to understand. Hence the manuscript needs thorough revision. For example, in the paragraph below Fig. 18, what does the sentence "The acceleration of Fig. 18 can reach Fig. 14, and ...." mean?
11. Fig. 23: The name of the axes start with "the ...". Please correct.
12. Conclusion, point (4): ".... this chapter ....", which chapter?
"The friction force is the lack of understanding of the friction properties." What does this sentence mean? There are a lot of similar sentences that are difficult to grasp.
13: The conclusion is too wordy. Focus on concluding remarks on the results of the study.
14. The numbering of the reference list is messy.
Author Response
Applied Sciences Editorial Office<applsci@mdpi.com>
Article: Fault Mechanism and Identification Study for Cam Mechanism of Clearance
Manuscript ID: Applsci-1761071
NUC., Taiyuan, Shanxi, China
19th June 2022
Dear reviewer,
It is the first appreciation of the advice of the respected anonymous reviewers. According to this advice and notes, the article manuscript has been revised. The following is in detail.
The change includes six sections. First is the background introduction second. It is in the first part. Then is the reference documentation; we omitted the references before 2006 and reminded those for 15 years, and renewed the reference section. The third part is an additional detail. They are on pages 1, 5, 16, 17, 22, 23, 25, and 26. Detailed locations are shown on the diagram introduction of the revised part22022.06.19 (DIRP, in the form of pdf). The revised part is dotted with red letters. The blue letters are the reviewer's questions (Q) and my answers (A). The black letters represented the main body of the article. In addition, all the changed marks have remained in the submitted doc fil. 90% of all figures have been renewed for the time limit. In addition, we also resort to the MDPI English language service.
Then this is an answer to the second reviewer.
Q1: At the beginning of the manuscript, I couldn't find the precise definition of the aim and the new scientific excellence of the current state of the art. Moreover, the introduction is chaotic and does not introduce the reader to the field of study.
A: Rewrite the background section, and review all the
References (DIRP, page1-3).
Q2: The list of abbreviations should be placed at the beginning of the manuscript. After the first read of the manuscript, I could not understand most of the used formulas. Additionally, some subscripts are missing in the manuscript and the list of used terms.
A: The full name is given at the first mention of a particular word, with an acronym form (page 1).
Q3: The quality of the time series is terrible, and it looks like a copy from the book. Please provide proper visualization of the results obtained.
A: Renew all the time-series figures.
Q4: In conclusion, I cannot find the next steps of your research.
A: Delete the fuzzy conclusion and summarize it. In the next step, I will try to study further the lubrication environment and degradation of the cam-roller mechanism of the five rollers
Q5: The list of cited papers is quite old. I believe that are pretty new papers related to your research, which in my opinion, should be included.
A: Delete all the references before 2006, only enumerate the relevant references in the past 15 years, and update the reference section.
Finally, thanks a lot for all the advice again. If any questions, please contact me at 20080005@nuc.edu.cn.
Best wishes to you.
Yours sincerely,
Xuefang Chang

Reviewer 3 Report
Dear Authors,
I found your article very interesting, but the manuscript requires a lots of changes to the present form before accepting it for publication. I hope that the list of following remarks will help the authors to improve the scientific quality of the paper:
1. At first, in the beginning of the manuscript I can’t find the precise definition of the aim and the new scientific excellence to the current state of the art. Moreover, the introduction is very chaotic and doesn’t introduce the reader to the field of study.
2. The list of abbreviations should be placed in the beginning of the manuscript. After the first read of the manuscript I couldn’t understand the most of used formulas. Additionally, some subscripts are missing, both in the manuscript and in the list of used terms.
3. The quality of time-series is really bad and it looks like a copy from the book. Please provide proper visualization of results obtained.
4. In conclusions, I can’t find the next steps of your research.
5. The list of cited papers is quite old. I believe, that are quite new papers related to your research, which in my opinion should be included.
After improving above described issues in the paper I’d like to give my positive opinion on signing my review report.
Author Response
Applied Sciences Editorial Office<applsci@mdpi.com>
Article: Fault Mechanism and Identification Study for Cam Mechanism of Clearance
Manuscript ID: Applsci-1761071
NUC., Taiyuan, Shanxi, China
19th June 2022
Dear Mr. Nikola Momic,
It is the first appreciation of the advice of the respected Nikola and the two anonymous reviewers. The two reviewers present the twelve pieces of advice, and the respected Nikola informs the five-note. According to this advice and notes, the article manuscript has been revised. The following is in detail.
The change includes six sections. First is the background introduction second. It is in the first part. Then is the reference documentation; we omitted the references before 2006 and reminded those for 15 years, and renewed the reference section. The third part is an additional detail. They are on pages 1, 5, 16, 17, 22, 23, 25, and 26. Detailed locations are shown on the diagram introduction of the revised part22022.06.19 (DIRP, in the form of pdf). The revised part is dotted with red letters. The blue letters are the reviewer's questions (Q) and my answers (A). The black letters represented the main body of the article. In addition, all the changed marks have remained in the submitted doc fil. 90% of all figures have been renewed for the time limit. In addition, we also resort to the MDPI English language service.
Next, this is an answer to the first reviewer. Please refer to the DIRP.
Q1: All the figures are blurry and need to be more explicit.
A: Due to the limited time, only 90 percent of the pictures have been updated, and if necessary, continue to update them later.
Q2: How the author describes the impact model between the fixed cam curve groove, roller follower, and the bolt at high speeds in the presence of clearance.
A: A cam roller mechanism model with a gap is given with the necessary text description. (DIRP, page 5)
Q3: What is the effect of pure rolling and sliding on the dynamic characteristic of the roller follower in the presence of clearance at high speed?
A: Because of the gap, the collision between the main roller and the two end surfaces often occurs. During an average cycle, the main roller meets the front end once, and the contact force of the main roller and the rear end surface changes periodically. The main roller cannot meet the conditions for pure rolling, so the rolling cannot occur, and translational motion can only do. The contact surface is a finite domain inside contact with sliding friction. (DIRP, page 17)
Q4: The author claimed that if there is no clearance, the dynamic motion of the system should be periodic. More explanation is needed. In the presence of clearance, the system will move with non-periodic motion and chaos.
A: Looking closely at Figs 14 and 15 to know this point. It can also be obtained from the law of numerical change. (DIRP, page 16)
Q5: In figures 17-21, I think the author studied the effect of acceleration against frequency using Fast Fourier Transform analysis, but he did not discuss the dominant frequency and fundamental frequency.
A: the dominant frequency is 16384 Hz, and the fundamental frequency is 512 Hz. The related signal processing is not the critical point of the article. We chose to simple introduction.
Q6: The author claimed that he used different sizes for the roller follower. Could you please explain the effect of these different rollers' sizes on the positive pressure?
A: In the following three situations of R=R0, R=R0-0.05, and R=R0-0.1, the changing trend of the contact force is roughly the same. However, at the same position, the positive pressure of the roller with a large radius corresponds to a slightly more tremendous positive pressure of the contact surface of the interface. The side pressure after colliding with the cam curve groove is a little smaller.
Q7: In the reference section, two numbers affect the citation. Please check the reference number (4) title, especially the word (Global). It should be (Globoidal). Which number did the author take?
A: Revised all the related references and contents.
Then this is an answer to the second reviewer.
Q1: At the beginning of the manuscript, I couldn't find the precise definition of the aim and the new scientific excellence of the current state of the art. Moreover, the introduction is chaotic and does not introduce the reader to the field of study.
A: Rewrite the background section, and review all the
References (DIRP, page1-3).
Q2: The list of abbreviations should be placed at the beginning of the manuscript. After the first read of the manuscript, I could not understand most of the used formulas. Additionally, some subscripts are missing in the manuscript and the list of used terms.
A: The full name is given at the first mention of a particular word, with an acronym form(page 1).
Q3: The quality of the time series is terrible, and it looks like a copy from the book. Please provide proper visualization of the results obtained.
A: Renew all the time-series figures.
Q4: In conclusion, I cannot find the next steps of your research.
A: Delete the fuzzy conclusion and summarize it. In the next step, I will try to study further the lubrication environment and degradation of the cam-roller mechanism of the five rollers
Q5: The list of cited papers is quite old. I believe that are pretty new papers related to your research, which in my opinion, should be included.
A: Delete all the references before 2006, only enumerate the relevant references in the past 15 years, and update the reference section.
Finally, thanks a lot for all the advice again. If any questions, please contact me at 20080005@nuc.edu.cn.
Best wishes to you.
Yours sincerely,
Xuefang Chang

Round 2
Reviewer 1 Report
I recommended this paper for publication in this journal pending minor revision, I have just one comment:
(1) The author claimed that if there will be no clearance, the dynamic motion of the system should be periodic. More explanation is needed. In the presence of clearance the system will move with non-periodic motion and chaos.
Author Response
Applied Sciences Editorial Office<applsci@mdpi.com>
Article: Fault Mechanism and Identification Study for Cam Mechanism of Clearance
Manuscript ID: Applsci-1761071
NUC., Taiyuan, Shanxi, China
7th July 2022
Dear reviewer,
I appreciate your second permission very much and feel there is a slight divergence.
It is about the periodic movement of the main roller. After the firing rate is stabilized, the automaton tends to the periodic motion. So the main roller with no clearance regular slides along the curve groove. With a significantly smaller gap, the main roller rolls along the curve groove and collides with the front and rear wall of the curve groove periodically. The main roller will move with non-periodic motion and chaos when the clearance is above a particular door value. But before the firing rate stabilized, the rules above did not exist. In the paper, if there is no emphasis, the condition of the stabilized firing rate is acquiescent. The related content of the article manuscript has been revised again.
If you have any questions, don't hesitate to contact me by email at 20080005@nuc.edu.cn. All matters related to the paper are welcome. Thank you very much.
Best wishes to you.
Yours sincerely,
Xuefang Chang

Reviewer 2 Report
The responses to my first concerns seem to be addressed. While reading the manuscript once again, I came across some other issues of concern that are listed below.
1. The title is not clear. Consider to modify to, for example as
"Study of Fault Mechanism Identification of Clearance in Cam Mechanism"
2. Line 18: "... is deduced in this paper." Check if the word "deduced" is proper for this sentence.
3. "dynamics" can be too general as a keyword for this paper.
4. Line 39. Bring the full stop behind the reference citation as "... obtained [1]. The idea ..." There are many similar appearances in the manuscript that need fixed.
5. Line 62: " ... , the researchers conducted ..." Which researchers? Citation is required.
6. Line 81: " ... cam roller follower was carried out." Who carried out? Reference is required.
7. Line 89: "... a significant role in aspect of the shot speed ..." --> "... a significant role with respect to the shot speed ..."
8. Figs 2 & 3 are not sufficiently described. Presenting these two figures under a single caption may improve the clarity in terms of the relation between them.
9. Line 154: " .... 27500N, and ..." --> " .... 27500 N, and ..." Please provide spacing between a value and its unit. Check and fix throughout the manuscript.
10. What are the horizontal and vertical arrows in Fig. 5, just below number 3 in the figure?
11. Fig. 6, Line 172: Why is "F - the friction force" is included in the note, while it does not appear in the figure?
12. Line 309 and 410: Move the "Note:" text to the figure (Fig. 9).
13: Line 318: The two sentence seem to be identical.
14: Fig. 10: I wonder if the figure is symmetrical. The given dimensions do not show that.
15: Line 412: "in Eq. 31." Do you mean Eq. 35?
16. Fig. 11: Please remove the red underlines before taking the screenshot or insert original image. The flowchart do require description.
17. Adjust the font size of Fig. 12, which is too large, while the rest, i.e. Fig. 13 and 14 are much smaller. Use standard abbreviated form of "second" in all figures with time axis, Fig. 12 - 17.
18. Line 587: "..., as shown from Fig. 22 to Fig. 17., and ..." --> "..., as shown in Fig. 17 to Fig. 22., and ..."
19. Fig. 27: How are the 3 states distinguished? Please correct the axis texts that start with "the number ...." and "the eigenvalue ..."
20. The conclusion does not reflect specific results or achievements.
21. The reference list requires reformatting according to the journal´s style.
Author Response
Applied Sciences Editorial Office<applsci@mdpi.com>
Article: Fault Mechanism and Identification Study for Cam Mechanism of Clearance
Manuscript ID: Applsci-1761071
NUC., Taiyuan, Shanxi, China
7th July 2022
Dear respected reviewer,
I appreciate your second permission very much. I also present a great admiration for your working attitude. According to your 21 suggestions, the article manuscript has been revised again. The following is in detail.
First, the paper title is changed to Study of Fault Identification of Clearance in Cam Mechanism. The second matter is including of the two contents improvement. On one hand, Figs 2 and 3 are put together and add the necessary words illustrate on pages 3 and 4. On the other hand, the conclusion at the end of the article is reorganized. The research finding includes two-part: one is the necessary condition of the roller group pure rolling of the multi-roller cam mechanism, which lays the foundation for the wear of the cam curve groove. The other is that the VK intelligent identification method can
effectively distinguish the motion distorted state of the cam mechanism from the normal state, which provides a reference for an insight into the wear mechanism of the complex mechanical system.
There are minor errors in the figures or words or the Literature citation format. I revised my paper according to your advice. After the MDPI language service, I yet check again and again. I marked the relevant parts as follows,
Q2: derived takes the place of deduced.
Q3: delete the dynamics and provide a new keyword: fault identification.
Q4:obtained change into achieved.
Q5:Xu et al.
Q6: [9-10].
Q7:with respect to is accepted.
Q9: the advice is adopted.
Q10: vertical arrow: y, horizontal arrow:z, as shown in figure 5.
Q11: F has been marked.
Q12: yes.
Q13:delete the first sentence: The critical point B is rolling.
Q14: The figure is not symmetrical. The figure was slightly modified to make the gap
more obvious.
Q15: It is Eq.35.
Q16: OK.
Q17: Adjust the font size of Fig.12.
Q18: yes, but it is 19-24.
Q19: New figures include three color lines.
Q21: The reference is edited again.
There are 21 pieces of advice in total. There are very timely and accurate. Thank you very much. Best wishes to you!
Yours sincerely,
Xuefang Chang

Reviewer 3 Report
The paper is only slightly corrected. I can see that results presented are not clear and are not presented in well manner. This is the main remark, which has to be revised. I don't recommend publishing the paper in the present form.
Author Response
Applied Sciences Editorial Office<applsci@mdpi.com>
Article: Fault Mechanism and Identification Study for Cam Mechanism of Clearance
Manuscript ID: Applsci-1761071
NUC., Taiyuan, Shanxi, China
7th July 2022
Dear respected reviewer,
I appreciate your second permission very much. I also present a great admiration for your working attitude. The conclusion at the end of the article is reorganized. The research finding includes two parts: the necessary condition of the roller group and pure rolling of the multi-roller cam mechanism, which lay the foundation for the wear of the cam curve groove. The other is that the VK intelligent identification method can effectively distinguish the motion distorted state of the cam mechanism from the normal state, which provides a reference for an insight into the wear mechanism of the complex mechanical system.
The part is changed into the following content:
This paper studies the gap between the main roller and the curve groove-the more the clearance, the stronger the front and rear motion of the main wheel. At the same time, the greater the shear force of the main roller shaft, the more likely the post is to break. In contrast, the smaller the gap, the less energy of the movement. The less the sheer force of the axis is, the better the protection of the axis is. However, in the case of too small clearance, the primary roller cannot rotate normally, and the higher performance of the lubricating oil also has been needed. This paper considers the gap between the roller and the curve groove and the direction of motion. Therefore, based on the requirement for pure rolling in the roller group, we can rapidly predict when the rolling wheel sticks to the front and rear wall and when it is in the middle. It provides favorable conditions for studying the wear of the cam curve groove of the automaton. The failure cam mechanism is researched given the kinematic and kinetic views of the multi-body system. Meanwhile, clearance and recoil are considered in deriving the pure rolling formula of the roller group. This paper focuses on the wear of the curved groove of the cam and the frequent braking and rapid reversal of the main roller caused by recoil. The VK algorithm proposed in this paper enables fast and accurate model identification. Based on dmkf2, which can effectively distinguish between the motion distortion state and the working state of the multi-roller cam mechanism, the present results provide a reference for an insight into the wear mechanism of the complex mechanical system.
The data transmission through online testing and monitoring systems can be achieved in future work. Not only can it timely obtain the related information on both the gap and the movement characteristics of the mechanism. Can it rapidly judge the current working conditions and quickly determine whether the roller or cam curve groove needs to be replaced. In future research, the degeneration of rollers with different strength materials can be studied in different lubrication environments to quickly replace the roller wheel, reduce the wear of the cam curve groove, save cost, and make new contributions to the development of the Gatling gun.
If you have any questions, don't hesitate to contact me by email: 20080005@nuc.edu.cn. All matters related to the paper are welcome. Thank you very much!
Best wishes to you.
Yours sincerely,
Xuefang Chang
